# Microbe-host interplay in atopic dermatitis and psoriasis

Nanna Fyhrquist (iD) et al.[#]

Despite recent advances in understanding microbial diversity in skin homeostasis, the relevance of microbial dysbiosis in inflammatory disease is poorly understood. Here we perform a comparative analysis of skin microbial communities coupled to global patterns of cutaneous gene expression in patients with atopic dermatitis or psoriasis. The skin microbiota is analysed by 16S amplicon or whole genome sequencing and the skin transcriptome by microarrays, followed by integration of the data layers. We find that atopic dermatitis and psoriasis can be classified by distinct microbes, which differ from healthy volunteers microbiome composition. Atopic dermatitis is dominated by a single microbe (*Staphylococcus aureus*), and associated with a disease relevant host transcriptomic signature enriched for skin barrier function, tryptophan metabolism and immune activation. In contrast, psoriasis is characterized by co-occurring communities of microbes with weak associations with disease related gene expression. Our work provides a basis for biomarker discovery and targeted therapies in skin dysbiosis.

Interactions between commensal or pathogenic microbes and the hosts they colonize are central to the maintenance of homeostasis and the initiation of disease[1]. This rapidly advancing field is now starting to bear the fruits of inter-disciplinary efforts but our understanding of microbe−host interactions is still limited[2].

Skin represents a primary tissue interface, where accessibility to both microbiome species and underlying tissue provides a unique opportunity for studies into host−microbiome interactions[3] in the initiation and maintenance of atopic/allergic or autoimmune-type inflammation[4,5]. Recent advances in analyzing microbial gene sequences in healthy skin has provided a comprehensive understanding of the classes of microbes and their diversity occupying distinct topographical niches[6]. Pioneering studies have expanded into shotgun sequencing approaches to further probe the taxonomic diversity and biogeography of microbes in small cohorts of healthy individuals[7]. Commensal skin microbes control adaptive skin immune homeostasis through interaction with specific subsets of antigen-presenting dendritic cells (DCs) and effector T-cell populations[8,9], supporting their own survival, and protecting against the overgrowth of pathogens. Insights into the association of distinct microbial classes with inflammatory skin disease and their impact on the host genome are just emerging. In atopic dermatitis (AD), as a model of atopic/allergic inflammatory disease, commensal skin microbes are associated with disease flares[10]. AD-related dysbiosis is frequently characterized by the colonization by *Staphylococcus aureus*, and simultaneous loss of other, potentially beneficial species. *S. aureus* colonizes skin effectively, and expresses several virulence factors[11] with proven roles in the pathogenesis of AD by studying effects in cell[12] and animal models[13,14]. *S. aureus* is associated with severity of the disease, and may be the result of a combination of detrimental effects from *S. aureus* on the one hand, and the loss of beneficial effects from other members of the skin microbiota on the other hand. Therefore, the use of antimicrobial therapeutics that target *S. aureus* may not be the optimal choice as they may also wipe out beneficial species or strains, and break mutualistic interactions between the skin and its microbiota. Psoriasis (PSO) appears to induce physiological changes at the lesion site, selecting for a specific microbiota. The psoriasis (PSO)-associated skin microbiome displays disease-specific features that might have a diagnostic value[15–17], but whether the differential microbiota has pathophysiologic significance remains undetermined.

Here, we present a large-scale, comprehensive analysis of the microbiome and microbiome-associated host transcriptome in skin of healthy volunteers (HV), AD, and PSO patients, revealing two distinct patterns of host−microbe interactions in chronic skin inflammation. We report a significant increase in the abundance of *S. aureus* and loss of anaerobic species in AD, while PSO is characterized by the co-occurrence of multiple organisms, including *Corynebacterium* and *Finegoldia* species. In AD lesions, which are abundantly colonized by *S. aureus*, toxin production and metabolic reprogramming are highly over-represented microbial functions. The host, in turn, responds to the changes in the microbiota via altered expression of genes related to barrier function, metabolic reprogramming, antimicrobial defense mechanisms and T helper type 2 ($T_H2$) signaling. In PSO, our results suggest that members of *Corynebacterium* may play a regulatory role, which is reduced in disease.

## Results

### The skin microbiotas in AD and PSO are highly distinct. Skin swabs were collected using a standardized protocol (Supplementary Fig. 1a−c). A total of 3.36 million 16S rRNA gene reads were analyzed using QIIME v1.8.0[18] resulting in the identification

of 17,725 operational taxonomic units (OTUs) at 99.3% identity level. After removal of rare OTUs, 3342 remained. Blasting sequences against the Greengenes 16S rRNA gene database revealed healthy skin microbiomes consistent with previous reports[6]. An overall analysis indicated clear differences between AD and HV (Fig. 1a, b, Supplementary Fig. 2), confirmed by nonmetric multidimensional scaling (NMDS) analysis of the 95 most abundant OTUs (Supplementary Fig. 3a, Supplementary Table 1). AD also showed a reduction in diversity (Supplementary Fig. 3b). A total of 51 highly abundant OTUs showed significant differences between HV, AD and PSO (Fig. 1a).

We additionally carried out a careful analysis of the influence of confounding factors, including age, anatomical location, gender and clinical center (Supplementary Table 2, Supplementary Fig. 3c). After correction for the confounding effects, 8 out of 17 confounded OTUs from AD and 5 of 13 OTUs from PSO retained significance. For example, *C. kroppenstedtii* was associated with age, *S. aureus* with anatomical location and *Lactobacillus sp.* with gender, but they all remained significantly associated with disease after correction ($p < 0.01$). Associations were tested using the Kruskal−Wallis test for body site and institution, the Mann−Whitney $U$ test for gender, and Spearman correlation for age.

The remaining 11 most significant OTUs are shown in Fig. 1c. The most significant result is an increase in the abundance of *S. aureus* in AD, associated with a significantly lower abundance of OTUs representing strictly anaerobic bacteria in AD (Fig. 1d). The loss of anaerobes in AD is not driven by *S. aureus*, indicated by repeated analysis of samples devoid of *S. aureus* (Supplementary Fig. 2d). Significant changes in PSO compared to HV include increases in the abundance of *C. simulans* and *C. kroppenstedtii* as well as *Finegoldia* and *Neisseriaceae* species. *Lactobacilli*, *Burkholderia spp.* and *P. acnes* were lower in abundance in both AD and PSO compared to healthy skin.

**Microbiota-based classification of AD and PSO**. We next asked whether the skin microbiome discriminates inflammatory skin pathologies. A supervised learning pipeline was employed to construct classifiers to explore the key sets of microbial taxa. We identified 26 microbes discriminating AD vs. HV cohorts with an area under the curve (AUC) of 0.94 (class errors HV = 0.03, AD = 0.27, Fig. 2a). The most discriminative taxa were of the genus *Staphylococcus*, including *S. aureus* ($Z = 14.0$), *S. epidermidis* ($Z = 5.8$), *Staphylococcus* spp. ($Z = -6.9$) and *Burkholderia* spp. ($Z = -7.5$). An NMDS analysis highlights the role of *S. aureus* in differentiating AD (red dots) from HV (blue dots; Supplementary Fig. 4a). We identified 24 microbes that differentiate PSO vs. HV with an AUC of 0.85 (class errors HV = 0.09, PSO = 0.33, Fig. 2b). The top discriminating microbes were *C. simulans* ($Z = 15.5$), *Neisseriaceae* g. spp. ($Z = 6.9$), *C. kroppenstedtii* ($Z = 5.5$), *Lactobacillus* spp. ($Z = -8.4$) and *Lactobacillus iners* ($Z = -3.5$). An NMDS analysis demonstrated a clear separation boundary between HV (green dots) and PSO (green dots; Supplementary Fig. 4b). We identified 15 microbes that differentiate between AD and PSO with AUC of 0.86 (class errors AD = 0.29, PSO = 0.13, Supplementary Fig. 4c). The top microbes were *S. aureus* ($Z = 14.0$), *S. epidermidis* ($Z = 3.9$), *Finegoldia* spp. ($Z = -6.5$), *C. simulans* ($Z = -5.0$), and *C. kroppenstedtii* ($Z = -4.4$). An NMDS analysis highlights the importance of *S. aureus* in AD (Supplementary Fig. 4d).

**Communities of microbes associate with AD or PSO**. To understand the interactions between communities of microbes under different disease states, we used network principles to express co-occurrence relationships. We found distinct

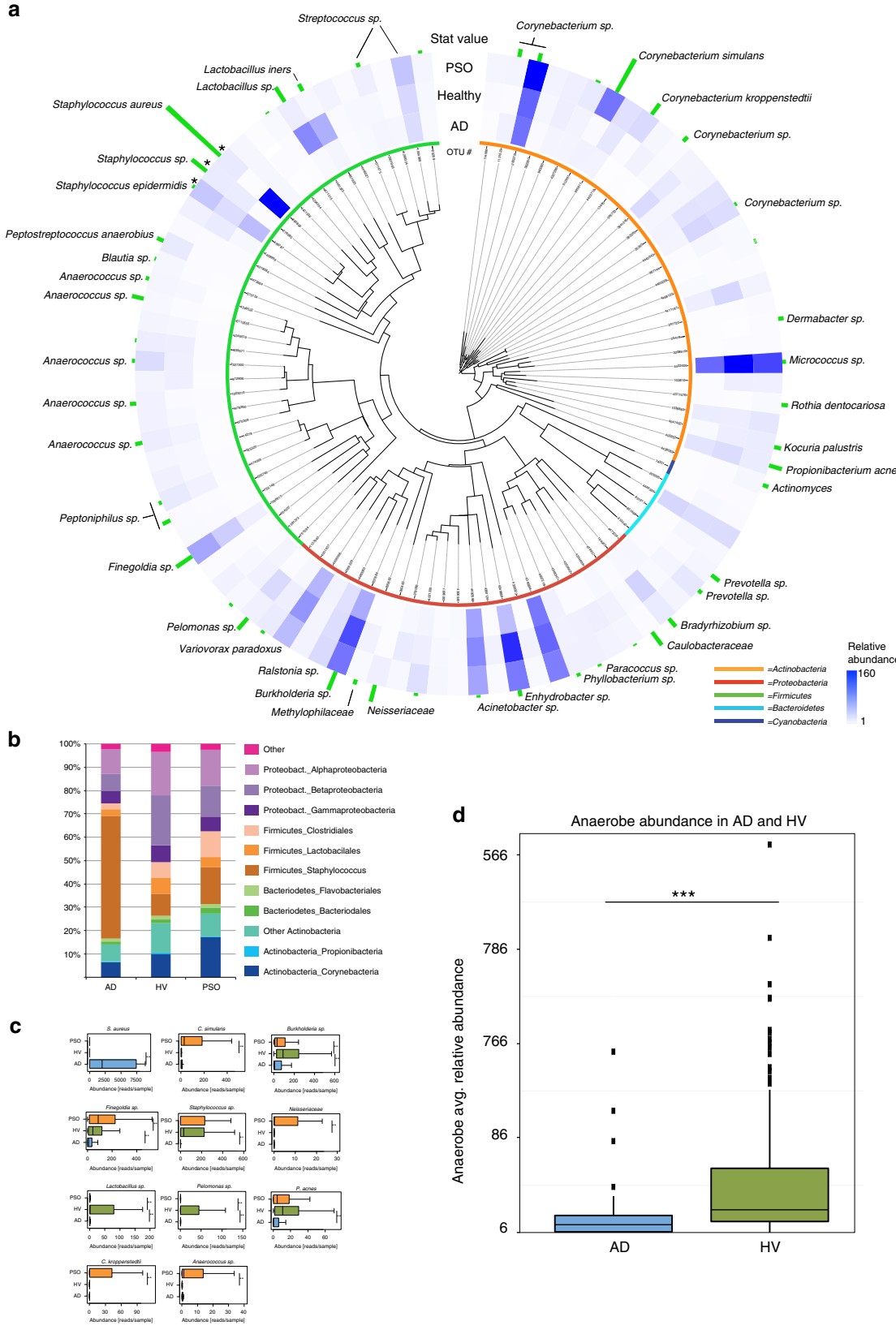

differences between the community structures of microbes associated with AD and PSO. For microbes associated with AD, SparCC[19] correlation between taxa resulted in 19 species with a correlation above a threshold of 0.2 ($p < 0.05$, Fig. 2c). *S. aureus* negatively correlated with species including *Corynebacterium*

spp., *S. epidermidis*, *Tepidimonas* spp. and *Phyllobacterium* spp. Of the 24 microbes identified as important for PSO classification (Fig. 2b), 16 species showed significant SparCC correlation (SparCC > 0.2, $p < 0.05$, Fig. 2d). The most discriminant taxa *C. simulans* and *C. kroppenstedtii* displayed positive correlations

**Fig. 1** Characterization of the skin microbiome in AD and PSO. The results show a typical range of skin microbiomes in HV ($n = 115$) and significant changes in AD ($n = 82$) and PSO ($n = 119$). **a** An evolutionary tree based on 16S rRNA gene sequences, abundance and statistical significance of the 95 most abundant OTUs. The blue color intensity in the heat map shows the relative abundance of each OTU. The three *Staphylococcus* OTUs (indicated by asterisks) were calculated using a wider scale, due to their relatively high abundance in certain samples. The length of the green vertical bars indicate nonparametric statistical score (Kruskal−Wallis test, FDR, $p < 0.05$). The color bars by each OTU number indicates bacterial phylum (green: *Firmicutes*; red: *Proteobacteria*; cyan: *Bacteroidetes*; blue: *Cyanobacteria*; orange: *Actinobacteria*). **b** The most abundant bacterial groups depicted for HV, AD and PSO. **c** Statistical analysis (Mann−Whitney *U* test (FDR, $p < 0.05$)) of the 11 OTUs showing the most significant changes in AD and/or PSO vs. HV after correction for confounding factors. The values on the *x* axis are in number of reads, out of a total of 8495 reads/sample. The asterisks indicate statistically significant differences and correspond to $p < 0.01$ (**) and $p < 0.001$ (***). The center line in the boxplots corresponds to the median, the bounding box is the interquantile range (IQR) and the whiskers are defined as 1.5 times IQR. **d** Statistical analysis (Mann−Whitney *U* test, $p < 0.05$) of the relative abundance of OTUs representing strictly anaerobic bacteria in AD lesions and HV. The *x*-axis units and the elements of the boxplots are as in (**c**). The source data files used to generate the present figure are available from the NCBI Sequence Read Archive under accession PRJNA554499

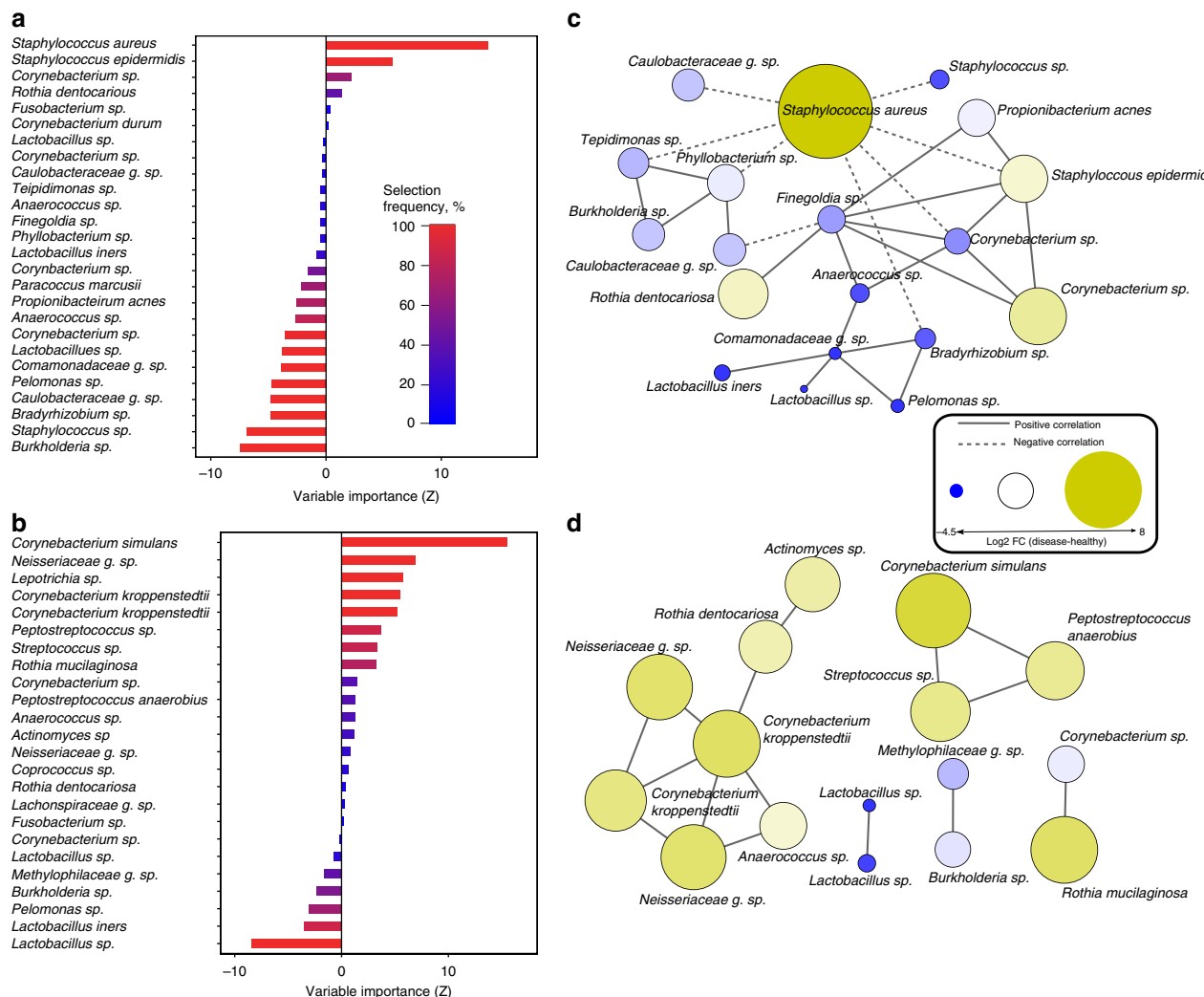

**Fig. 2** Microbiota-based classification of AD and PSO. **a** Variable importance for the best set of discriminatory AD ($n = 82$) taxa identified through Random Forest feature selection and classification (RF) analysis. Bars are colored by selection frequency (Red = OTU selected in all folds, blue = OTU selected in one fold). **b** Variable importance for the best set of discriminatory PSO ($n = 119$) taxa identified by RF analysis. **c** Co-occurrence network of AD-associated OTUs selected by RF selection. Pairwise correlations were calculated over AD samples using SparCC and OTUs with a mean $Z > 0.2$ are shown. A connection represents a correlation $>0.2$ and significant $p < 0.05$ correlation. Solid and dashed lines respectively represent positive and negative correlations. The size and color of each node is proportional to the log2 fold change between healthy and disease. **d** Correlation network of PSO-associated OTUs selected by RF. The source data files used to generate the present figure are available from the NCBI Sequence Read Archive under accession PRJNA554499

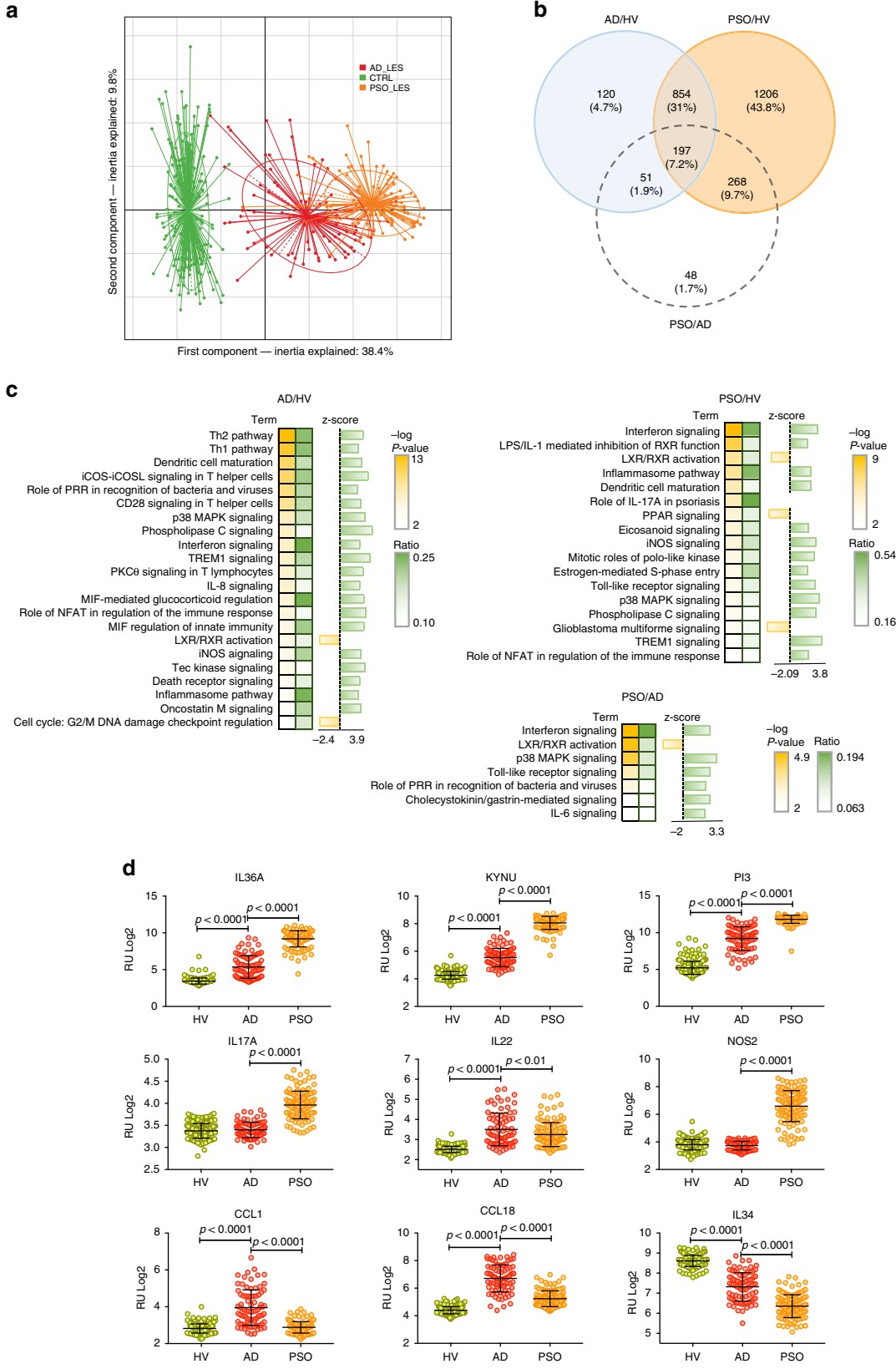

with *Streptococcus* spp., *P. anaerobius* and, *Anaerococcus* spp., *Neisseriaceae* g. spp., and *Rothia dentocariosa* respectively. Comparison across microbial interactions in PSO indicate that rather than a single species dominating the microbial landscape (as in AD), we observe multiple species associated with this disease type.

**Overlaps and distinctions of AD and PSO transcriptomes**. The transcriptomes of full-thickness skin biopsies were defined based on a stringent $10^{-5}$ FDR level and a fold change (FCH) of 1.5, or greater. A principal component analysis (PCA) revealed a clear separation between HV, AD, and PSO (Fig. 3a). Differential gene expression analysis identified 1232 genes differing between AD

**Fig. 3** The AD and PSO skin transcriptomes. **a** Projection of AD (red, $n = 82$), PSO (orange, $n = 119$) and HV (green, $n = 115$) transcriptome profiles in the subspace spanned by the two first components of the principal component analysis (PCA) performed on the 1000 most variant genes. **b** Venn diagram of the differentially expressed genes in the AD vs. HV, PSO vs. HV and PSO vs. AD contrasts (identified using the *Limma* linear model and empirical Bayes method, cut-off: log2 FC > 0.58 and FDR $p < 1 \times 10^{-5}$, corrected using the Benjamini−Hochberg method). **c** IPA canonic pathway analysis of significantly enriched functions in AD and PSO gene signatures and the PSO vs. AD contrast. **d** Statistical analysis (unpaired $t$ test) of selected top up- and downregulated genes. The center line in the dot plots corresponds to the mean, and the error bars to the standard deviation. The source data files used to generate the present figure are available from EBI ArrayExpress under accession E-MTAB-8149

and HV, and 2525 between PSO and HV, and 1051 genes shared between AD and PSO (Fig. 3b).

For functional insight into the disease-specific genes, we performed Ingenuity Pathway Analysis (IPA), revealing significant overrepresentation of $T_H2$ and $T_H1$ signaling, dendritic cell maturation and iCOS-iCOSL signaling in helper T cells in AD, and Interferon signaling, LPS-IL-1-mediated inhibition of RXR function, the Inflammasome pathway and Th17 signaling in PSO. The levels of Interferon signaling and p38 MAPK signaling distinguished PSO from AD (Fig. 3c).

*TNF* and *IFNG* were identified as upstream regulators in both AD and PSO, whereas *IL4* and *IL13* emerged as unique features in AD, underpinning the central role of $T_H2$ -associated signaling in AD (Supplementary Fig. 5b). Gene Ontology (GO) enrichment analysis highlighted chemotaxis, inflammatory response, and extracellular matrix organization in AD, and leukocyte activation in both AD and PSO (Supplementary Fig. 6). Among top upregulated genes in AD and PSO we observed inflammatory mediators (*S100* proteins, defensins, matrix metalloproteinases, IL-1 family cytokines), T helper-related genes (*CCL1*, *CCL18*, *IL17A*, *IL22*, *PI3*/Elafin), barrier genes (*KRT16*, *SERPINB4*, *KLK9*, *FLG2*, *LCE5A*, *CLDN8*), and genes involved in tryptophan (trp) metabolism (*KYNU*). Top downregulated genes included *IL34*, anti-inflammatory *IL37*, and *NOS2* (Fig. 3d, Supplementary Fig. 7).

**The *S. aureus*-induced host gene signature in AD**. Taking advantage of our large microbiome and transcriptome datasets, we addressed the interplay between host and microbes. We stratified AD samples into "high" and "low" groups, based on microbial abundance, including top ($n = 27$) and bottom ($n = 25$) tertiles in the analysis. The low abundance samples were devoid of *S. aureus*, whereas high abundance samples exhibited in all cases a high abundance of *S. aureus* (87–99%).

First, we explored the microbial gene repertoire of *S. aureus* high and low groups. Using PiCRUST that predicts functional content based on the *16S* rRNA gene marker, we observed the enrichment of bacterial toxins, the two-component system and glycolysis in the *S. aureus* high group compared with the low group (Supplementary Fig. 8a−b). The majority of the functions were contributed by *S. aureus* OTUs (Supplementary Fig. 8c). To validate OTU-based predictions, we performed whole genome sequencing (WGS) on a limited sample set, showing close agreement between the two independent sequencing methods (Supplementary Fig. 9). WGS data confirmed the enrichment of major bacterial toxins, hld and plc (alpha- and delta-toxin, respectively), and genes associated with galactose metabolism, the phosphotransferase system, the two-component system, and glycolysis and gluconeogenesis, in the *S. aureus* high samples (Supplementary Fig. 8c).

Next, we investigated gene expression profiles in the underlying skin. Comparison of the transcriptomes between *S. aureus* high and low samples revealed a set of 256 significant genes (FDR < 0.05, FCH ≥ 1.5) (Supplementary Fig. 10a, Supplementary Table 3). To explore whether the *S. aureus*-regulated genes were relevant to global features of AD pathophysiology, we created a

co-expression network based on pairwise Pearson correlation ($r > 0.7$), using all significant AD-associated genes. Network community detection[20] identified ten distinct modules, enriched for disease-relevant functions (Fig.4a, Supplementary Fig. 10c). Projecting *S. aureus*-regulated genes onto the AD network revealed significant enrichment in genes that mapped to modules M1 and M5 (hypergeometric test, FDR < 0.05), associated with keratinocyte differentiation, and extracellular matrix organization, respectively (Fig. 4b).

Functional analysis of the *S. aureus*-regulated genes using GO and IPA revealed the enrichment of keratinization and skin development (Fig. 4c), and $T_H17$ signaling and tryptophan (trp) degradation (Supplementary Fig. 11a), respectively. IPA predicted *IL1B*, *TNF* and *IFNG* as top upstream regulators (Fig. 4d) and leukocyte migration and development of epithelial tissue as downstream effects (Fig. 4e). While upregulated genes were enriched for inflammatory signaling (Supplementary Fig. 11b), downregulated genes were over-represented for mainly skin development (Supplementary Fig. 11c). Top genes included skin barrier and antimicrobial factors (*S100A7*, *DEFB4A/B*, *S100A9*, *MMP12*, *FLG2*, *CLDN8*, *ADAM12*), components of trp metabolism (*KYNU*, *TDO2*, *KMO*), immune activation (*IL1B*, *CCL2*, *CCL19*), and $T_H2$ signaling (*IL4R*, *IL5*, *IL13*, *PI3*, *TNFRSF4*, *CCR4*). Moreover, *HIF1A* and its targets *HK2* and *PFKP* were among the significantly regulated genes (Fig. 4e, f, Supplementary Fig. 10b, d−e).

Since the kynurenine pathway was enriched in the *S. aureus* "high" samples, we performed transcriptomic reconstruction of trp breakdown, indicating the accumulation of 3-hydroxyanthralic acid (3-HAA) in the skin (Supplementary Fig. 12a). Further, we investigated to what extent AD-associated strains of *S. aureus* may depend on trp, and isolated 32 *S. aureus* strains from AD patients with moderate to severe manifestation of the disease. We found that 66% of the isolated strains grew independent of trp (Supplementary Fig. 12b, Supplementary Table 4), and this observation was further supported by our WGS data (73% trp biosynthesis-related genes in the *S. aureus* high samples) (Supplementary Fig. 12c).

Finally, to explore the impact of *S. aureus* on gene targets identified by this study, organotypic human epidermal equivalents were topically exposed to $10^6$ CFU *S. aureus* for 24 h, followed by measurement of gene expression by qPCR (Supplementary Fig. 13a−c). *S. aureus* significantly induced the expression of *DEFB4*, *PI3*, *IL4R* and *S100A9* (Supplementary Fig. 13d).

**Integrative analysis of the PSO microbiome and transcriptome**. Next, we combined the PSO microbiome with associated skin transcriptomes. Testing the top discriminating microbes in PSO in terms of their ability to partition the skin transcriptome did not yield significant results. Therefore, we constructed a co-expression network based on PSO differentially expressed genes, and partitioned it into 12 modules as described above[20] (Supplementary Fig. 14a). We identified associations between the top 25 most differentially abundant taxa between PSO and HV, and module eigengenes derived from the PSO co-expression

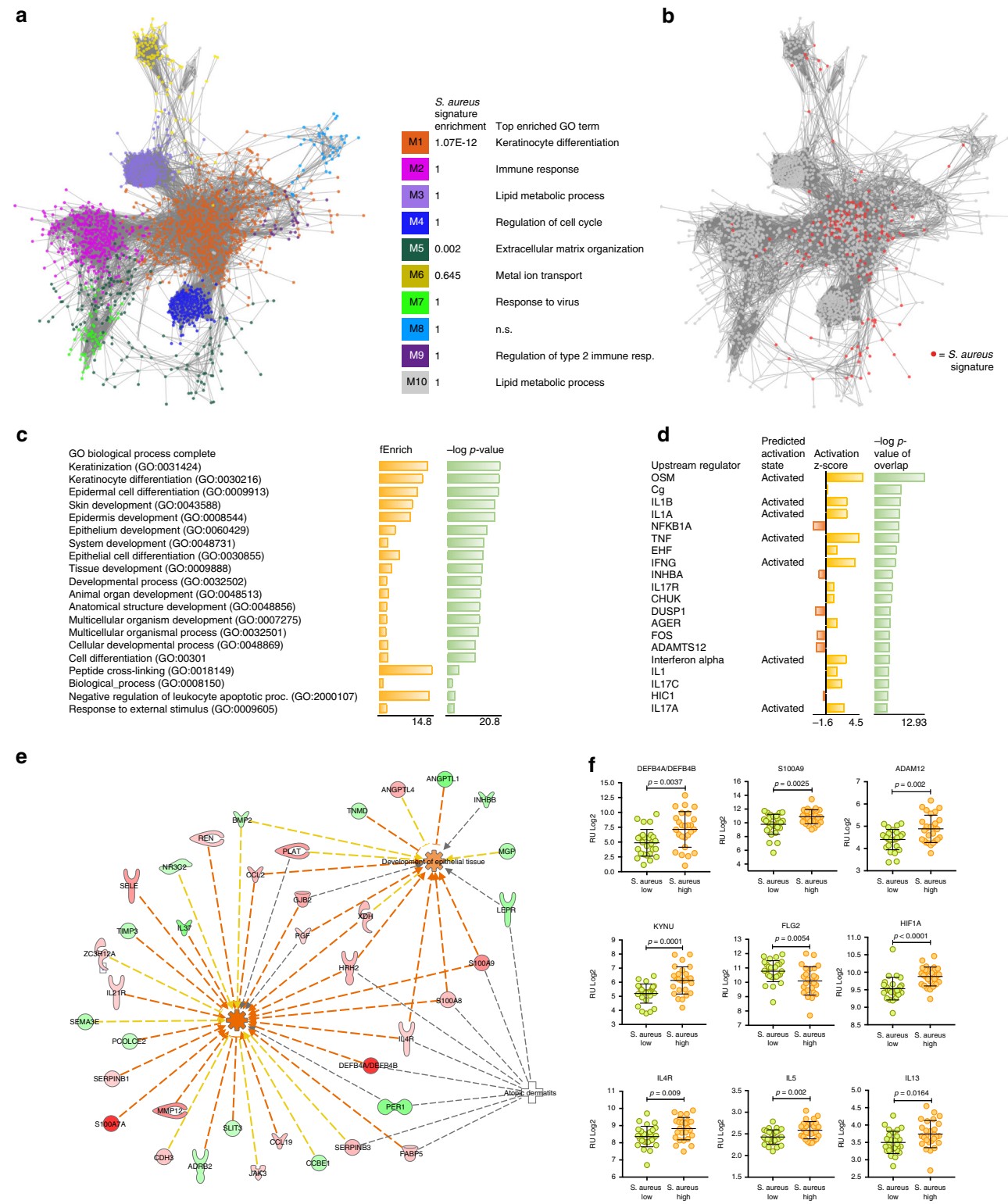

network, using linear models implemented in the MaAsLin package[21]. A linear model controlling for body site, institution, age and gender effects was fit for each microbe−module eigen-gene pair. Overall, six associations were identified (FDR < 0.20) (Supplementary Fig. 14b). Negative associations were identified between *Corynebacterium* spp. and three co-expression modules; M2, M3 and M10. M3 was strongly associated with the cell cycle, whereas M2 and M10 were enriched for inflammatory pathways including interferon signaling, and $T_H1$ and $T_H2$ activation

(Supplementary Fig. 14c). These results suggest that *Corynebacterium* spp. may play a regulatory role which is reduced in disease however. Overall, our results indicate that microbe−host associations in PSO are considerably less well defined than observed in AD.

**Microbiome associates with clinical severity in AD**. To inves-tigate the effect of microbiota and gene expression on clinical

**Fig. 4** The "*S. aureus* signature" and functional associations. **a** Using AD-associated genes (AD patients, $n = 82$) identified at FDR level $10^{-5}$, we created an AD gene co-expression network, which was partitioned into modules by network community detection. Top enriched GO terms are indicated for each network module. **b** Differential analysis between *S. aureus* "high" ($n = 27$) and "low" ($n = 25$) samples revealed 256 differentially expressed genes (FDR < 0.05, FCH ≥ 1.5). Hypergeometric tests revealed significant enrichment in *S. aureus*-associated genes (colored red) that mapped to modules M1 and M5. Gene annotations and Ingenuity Pathway analysis identified **c** enriched gene ontology terms in the *S. aureus* signature, and **d** predicted upstream regulators, respectively. **e** Molecular networks generated between top functions and associated genes. The red color of the gene symbols indicates upregulated genes, green indicates downregulated genes. Red colored edges indicate predicted activation, yellow edges indicate inconsistent findings, and gray edges lack a predicted effect. **f** Statistical analysis (unpaired *t* test) of RNA expression levels of selected genes in *S. aureus* "high" and "low" abundance samples. The center line in the dot plots corresponds to the mean, and the error bars correspond to the standard deviation. The source data files used to generate the present figure are available from the NCBI Sequence Read Archive under accession PRJNA554499, and from EBI ArrayExpress under accession E-MTAB-8149

severity, we performed feature selection coupled with multivariate regression. Genes and microbes were ranked according to their correlation with SCORAD or PASI indices and then regression models were trained onto the top $N$ features ($N = 5$–50 in increments of five), and the best performing model was selected. Prediction of SCORAD from microbial abundance was optimal with the top 35 species with Random Forest regression (Supplementary Fig. 15a). The accuracy was relatively modest (MAE = 12.35, correlation true vs. predicted = 0.55) suggesting that variability in microbial abundance and clinical severity are likely explained by other factors. Amongst the most highly correlated species were *Tepidimonas* spp., *Propionibacterium acnes*, and *S. aureus* (Supplementary Fig. 15b). Most of the selected species were negatively correlated with the exception of *S. aureus*, indicating that microbial diversity may be inversely associated with clinical severity in AD. Diversity correlation with SCORAD revealed a weak but significant correlation (cor = −0.27, $p = 0.01$).

Transcriptomic data outperformed microbial abundance for predicting SCORAD. The top model identified 15 genes using linear regression as the best set of predictive genes (Supplementary Fig. 15c), (MAE = 9.84, correlation true vs. predicted = 0.66). Amongst the top genes were *SEMA3D, IGSF10, LGR5* and *CASP10* (Supplementary Fig. 15d). No associations between clinical severity and microbial abundance in PSO were identified.

Finally, oozing and crusting in the AD lesions indicative of infection, correlated significantly with the abundance of *S. aureus*, supporting the link between colonization by *S. aureus* and clinical severity in AD (Supplementary Fig. 16).

## Discussion

Microbe−host interactions may be central to skin homeostasis, and dysbiosis may drive disease[10,15]. However, to what extent the skin microbiota may associate with the host skin phenotype and underlying mechanisms, remains elusive. To achieve a better understanding of the dialogue between the skin and the skin microbiome, we used skin as a model and AD and PSO as proxies for $T_H2$-associated atopic and $T_H17$-associated autoimmune inflammation, respectively, giving us the opportunity to compare two types of inflammation and the associated microbiomes and transcriptomes. Here we present the findings of a large cohort ($n = 316$) combining the analysis of skin microbiomes and associated transcriptomes, identifying unique gene profiles that characterize healthy vs. inflamed skin. We show that while AD is dominated by one single microbial species, multiple species associate with PSO, and the abundance of the dominating species in AD, *S. aureus*, correlates with disease relevant gene expression (overview of the main findings in Fig. 5).

Analysis of the two patient groups shows distinct differences between microbiomes. The most significant change, the increase in *S. aureus* abundance in AD, is not present in all samples. A

number of AD lesions contain little or no *S. aureus*, representing potentially different endotypes of the disease which deserve further investigation. The changes associated with psoriasis are more complex than for AD and involve many different bacteria, including *Corynebacteria* and *Finegoldia*. Through exploring classification models as a way to discriminate disease state, several of the OTUs were also strong predictors of disease status. *Lactobacillus* appeared to be consistently depleted in both diseases, whereas disease-specific species such as *S. aureus* in AD and *C. simulans* and *C. kroppenstedtii* in PSO were important for prediction in disease. Moreover, we identified two distinct community patterns by means of co-occurrence analysis. In AD, it was clear that *S. aureus* dominated the microbial landscape and negatively correlated with several skin commensals, such as *S. epidermidis* and *Corynebacterium* spp, and thus may be associated with the depletion of potentially regulatory or protective microbes. Importantly, recent studies have shown that *S. epidermidis* may specifically limit the growth of *S. aureus*[22], and disease severity is inversely correlated with the abundance of *S. epidermidis* relative to *S. aureus*[14]. In contrast to the dynamics revealed in AD, the PSO-associated interaction network suggested that multiple, co-occurring species, is a more representative model. Previous studies have shown that while single taxa are unable to discriminate between PSO and healthy skin, the combined relative abundances of selected genera (*Corynebacterium*, *Streptococcus* and *Staphylococcus*) attain significance across the groups[15]. In our hands, the same genera display clear patterns of co-occurrence, increasing in relative abundance in PSO lesions compared with healthy skin.

We observed a remarkable loss of strictly anaerobic bacteria in AD, indicating a switch in the skin microbiome from anaerobic to aerobic metabolism. Healthy skin is normally $O_2$ deprived[23], but a dry flaky skin and an impaired epidermal barrier function—characteristics seen in AD[24], may increase oxygenation and select for the low abundance of strictly anaerobic bacteria such as *Lactobacillus* spp or *Finegoldia spp*. At anaerobic conditions bacteria are fermenting organic matter, e.g. in skin the amino acid serine originating from filaggrin degradation, forming in particular lactic acid, propionic acid and other short chain fatty acids (SCFA). These metabolites lower the skin pH to pH < 5.5, maintaining a protective acidity of the skin. Potassium lactate is also one of the most relevant "natural moisturizing factors" of healthy skin. Furthermore, gram-positive anaerobe cocci such as *Finegoldia*, *Anaerococcus*, and *Peptoniphilus*, stimulate rapid induction of antimicrobial peptides response in human keratinocytes, which could be an important signaling mechanism to the keratinocytes when the skin is injured[25]. In the complete or partial absence of these organisms, danger signaling in keratinocytes and other barrier functions could be impaired, potentially favoring colonization by *S. aureus*.

We identified AD and PSO transcriptomes which overlap substantially with previously published studies[26–29]. While

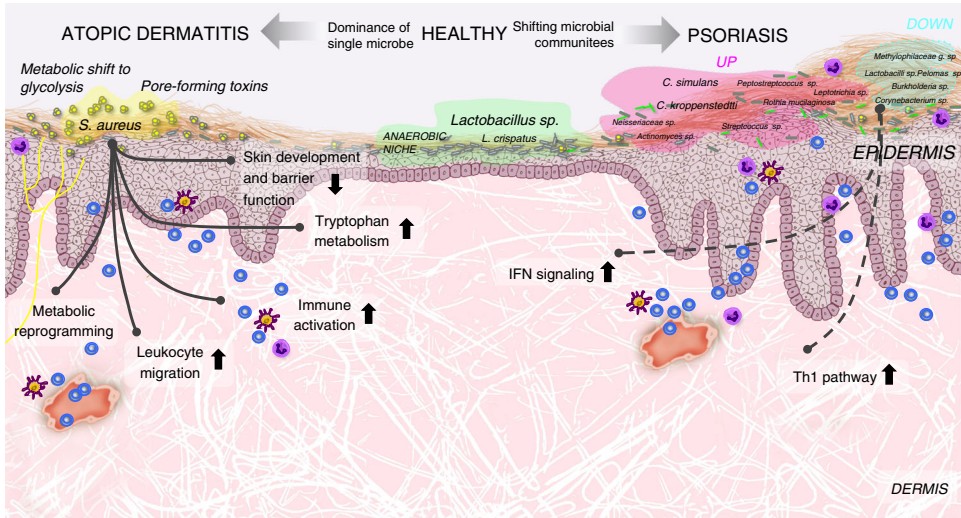

**Fig. 5** Host—microbe interaction in atopic dermatitis and psoriasis. AD is characterized by overgrowth of *S. aureus*, and loss of microbial diversity. In AD, colonization of the skin by *S. aureus* is associated with dysregulation of genes involved in epithelial barrier function, immune activation, leukocyte migration, trp degradation and metabolic reprogramming. PSO is associated with multiple species, including increased colonization by *C. simulans* and *C. kroppenstedtii*, and a loss of *Lactobacillus*, *P. acnes* and *Corynebacterium* spp., which may play regulatory roles

transcriptomic changes in PSO were dramatic, microbial changes were rather small. Yet, the opposite applied to AD lesions, suggesting a nonlinear relationship between the skin microbiota and the host transcriptome. Instead, the communication between host and microbiota might depend on defined subsets of genes, and the interaction is likely bi-directional, with the microbiota influencing host gene expression, which in turn forms a habitat for specific microbes.

Previous studies have identified colonization by *S. aureus* in AD; however, little is known about its potential mechanistic impact[10,13]. We identified differentially regulated host genes between AD lesional skin samples containing high or low abundance of *S. aureus*, functionally enriched for mainly three activities: skin barrier function, immune activation and trp metabolism. The *S. aureus* high associated microbiome was overrepresented by bacterial toxins (alpha-toxin and delta-toxin), the two-component system and glucose metabolism, essentially contributed by *S. aureus* specific OTUs. The highly adaptable and potent pathogen *S. aureus* is known to express a multitude of virulence factors, including toxins that impact on the skin barrier and the immune system. The pore forming alpha-toxin, for instance, which was highly enriched in the *S. aureus* "high" samples, likely plays a key role in the disruption of the skin barrier in AD patients, and through being able to activate the inflammasome, resulting in the secretion of *IL-1beta*, alpha-toxin is able to promote further inflammation[30,31]. Delta-toxin, in turn, is known to contribute to AD-associated pathology through the induction of mast cell degranulation and Th2 differentiation[13].

The host responded to *S. aureus* through upregulating beta-defensins (eg. *DEFB4*), and the expression of S100 protein family members (*S100A8*, *S100A9*, *S100A7*). Beta-defensins are potent antimicrobial peptides, contributing directly to local immune responses as chemoattractants for leukocytes, and through activating antigen-presenting cells (APCs)[32]. Psoriasin (*S100A7*) preferentially kills *Escherichia coli*, but has also bactericidal activity against *S. aureus*[33]. Being upregulated by proinflammatory cytokines, psoriasin functions as a T cell and neutrophil chemotactic agent. *S100A8* and binding partner *S100A9* are, in turn, expressed and released by activated phagocytes, and have powerful antimicrobial activities through the sequestration of essential trace elements. The *S100A8/A9* complex also protects

the host from infection by triggering toll-like receptor 4 (TLR4) and receptor for advanced glycation end-products (RAGE)-mediated inflammatory pathways, and through the recruitment of neutrophils[34,35]. Thus, besides targeting microorganisms, the antimicrobial defense response promotes inflammation, possibly feeding additionally into AD pathology.

Further, samples abundantly colonized by *S. aureus*, displayed significant changes in the expression of barrier genes, such as *CLDN8*, a component of tight junctions (TJs). TJs are key to epithelial barriers, reside immediately under the stratum corneum, and regulate the passage of water, ions and solutes through the skin. TJ defects, including dysregulated *CLDN8*, are associated with AD[36,37]. Moreover, the expression of metalloproteinases *ADAM12* and *MMP12*, which are implicated in tissue remodeling, cytokine and growth factor shedding, cell migration and adhesion[38], was modified. Alpha-toxin is known to specifically interact *ADAM10*, leading to the cleavage of cadherins in epithelial cell tight-junctions, and resulting in the destruction of cell—cell contacts[39], and consequently, the expression of ADAM10 was modified in *S. aureus* high samples.

Among proinflammatory factors that were influenced by *S. aureus*, we note dysregulation of *IL1B*, *CCL2*, *CCL19* and members of the Th2 signaling pathway. *IL1B* is a central mediator of inflammation[40], affecting all cells of the innate immune system and playing a key role in the differentiation and function of adaptive lymphoid cells. The chemokine *CCL2*, which is expressed mainly in basal keratinocytes, acts through attracting monocytes, dendritic cells, Th1 and Th2 cells[41], and the chemokine *CCL19* is involved in the colocalization of *CCR7*-positive dendritic cells with *CCR7*-positive T cells[42], facilitating antigen presentation and activation of T cells. Among Th2 associated genes, we observed the induction of chemokine receptor *CCR4*, cytokines *IL5* and *IL13*, cytokine receptor *IL4R*, and TNFRSR4, all central mediators of allergic pathomechanisms. Finally, the AD-associated peptidase inhibitor 3 *PI3*/elafin, a product of Th17 activation and implicated in Th2 differentiation[43], was significantly modulated in *S. aureus* high samples.

Colonization of the skin by *S. aureus* activates defense and tissue repair responses, which cause increased metabolic demands, requiring the tissue to switch to glycolysis[44]. When densely present, *S. aureus* may generate localized hypoxia and

biofilms[45], triggering hypoxia inducible factor (HIF) transcription factors, which drive the metabolic changes and shape the immunological response. In the *S. aureus* high samples, we observed significant upregulation of *HIF1A*, and its target genes *HK2*, *PFKP* and *IL1B*. *HK2* and PFKP are key regulatory enzymes in glucose metabolism. While glycolysis in the host is necessary to combat infection, a recent study shows that staphylococcal glycolysis is equally necessary to cause infection[44]. In line, we observed increased levels of glucose and galactose metabolism and glucose transport, in the *S. aureus* high associated microbiome, suggesting enhanced ability to compete with the skin for limited oxygen and glucose. Staphylococci also express hypoxia activated genes[46], some of them which are part of the two-component system, regulating metabolic activity in anaerobic conditions[44]. Accordingly, the two-component system was significantly enriched in the *S. aureus* high samples, including genes which facilitate anaerobic respiration and biofilm formation[47,48].

The essential amino acid trp and its catabolites are emerging as important mediators of host−microbe interactions, with multiple effects on host physiology on the one hand, and consequences for the survival of the microorganisms, on the other hand. Recent studies have shown attenuated trp metabolic pathways in the skin microbiota of AD patients, and protective functions of microbial trp catabolites in AD[49]. In turn, the host may use trp "depletion", or "starvation" as an antimicrobial strategy[50]. We generated a transcriptomic reconstruction of trp breakdown in atopic inflammation (Supplementary Fig. 13a), and detailed analysis of the pathway indicated significant upregulation of *TDO2*, *KMO* and *KYNU*, while alternative endpoints of the pathway were not regulated. Hence, the result suggests accumulation of 3-hydroxyanthranilic acid (3-HAA), and the pathway being locked into the generation of this metabolite. Interestingly, 3-HAA is considered an inflammatory mediator, and has been shown to interact directly with neuronal cells[51]. The depletion of key nutrients by the host is thought to be a mode of host defense during bacterial colonization, and certain microbes, including *staphylococci*, are sensitive to the depletion of trp by the host[50]. While investigating whether such mechanisms are at play in AD during *S. aureus* colonization, we found that the majority of *S. aureus* strains in AD patients were independent of supplied trp.

Taken together, our findings in AD patients imply that *S. aureus* may actively modulate the skin barrier function, while triggering an inflammatory response, including Th2 type signaling, potentially interfering with the regulation of *S. aureus* colonization and perhaps contributing to the exacerbation of clinical symptoms. Moreover, dense colonization by *S. aureus* may cause local hypoxia, leading to metabolic reprogramming of the host and its microbiota, and amplification of the inflammatory response. Our findings also suggest that *S. aureus* promotes trp degradation and the accumulation of proinflammatory metabolites in the host, which may help the host to maintain a defensive immune barrier, but may also feed into AD pathology. Finally, AD patients may be preferentially colonized by *S. aureus* strains that are capable of trp biosynthesis, providing them with a colonization advantage in the atopic environment. However, the defense mechanism of limiting trp may also result in adverse effects, i.e. repressing colonization by benign commensal bacteria, creating more dysbiosis.

Unlike in AD, where one species, *S. aureus*, was identified as the dominant microbe, we observed that multiple species demonstrated increased abundance in PSO. To establish a relationship between host transcriptomic profiles and microbial abundances, we screened PSO-associated microbes against modules of co-expressed genes, but detected only weak associations between potential pathogens and the expression of host transcripts. Several modules were found to be associated with

microbial abundance, four of which were associated with one particular taxon corresponding to *Corynebacterium* spp. This species was negatively associated with a module enriched for "interferon signaling", suggesting a potential protective, or regulatory role of pathways relating to psoriatic inflammation. This is of particular importance, since interferon signaling represents a key event during the initiation of psoriatic inflammation[52]. However, since the identified correlations with bacteria were weak, further studies are warranted for unraveling the potential role of fungi or viruses in psoriatic inflammation.

Finally, we applied regression algorithms to highlight potential links between transcriptomic and microbial biomarkers with disease severity. In AD a set of 35 microbes was identified and linked to clinical severity. Of these species, all, except *S. aureus*, were negatively correlated with disease severity. The association between the abundance of *S. aureus* and severity has been observed in previous studies[10,53], and specific strains of *S. aureus* contribute to a varying degree to severity, adding to the complexity of AD disease[14]. We lack *S. aureus* strain information in this study, but identify genes which appear to be driven by *S. aureus*. More than half of these genes are relevant to AD pathophysiology, overlapping with the AD transcriptome, but only a handful of these genes are associated with disease severity, suggesting that the *S. aureus* signature is driven mainly by *S. aureus*, not by severity. Lastly, we conclude that the host transcriptome predicted patient SCORAD to greater accuracy than the microbiome, suggesting that gene expression is a better predictor of clinical severity.

This study represents a large cohort with a rich dataset giving an opportunity for further detailed analysis of specific microbe−host interaction. For additional insight, full genetic diversity of the microorganisms need to be explored, adding information concerning strains and the functional potential, as well as information related to yet unexplored microorganisms including fungi, viruses and archaea. Principles of microbe−host interactions will need to be added to the concept of homeostasis and disease, and only an integrated approach will be able to follow the complex ecology of human health, paving the way for medical interventions that aim at preserving health-associated homeostasis between humans and their microbiota.

## Methods

**Subject recruitment and sampling**. Adult patients (18–83 years) with moderate-to-severe chronic AD (SCORAD score > 25, $n = 91$) and plaque-type PSO (PASI score > 7, $n = 134$) as well as healthy volunteers ($n = 126$) were recruited for the study (Supplementary Fig. 1). Microbiome samples and skin biopsies were obtained from areas with active disease in the upper back or posterior thigh in AD patients, and from the upper and lower back in PSO patients. Healthy volunteers were sampled in corresponding skin areas.

Each subject underwent a physical examination by a dermatologist and the medical history was recorded. The diagnoses were made by a dermatologist based on clinical presentation, personal history, laboratory findings and the criteria of Hanifin and Rajka[54]. The exclusion criteria included concomitant autoimmune diseases (e.g. rheumatoid arthritis, diabetes, alopecia areata, etc.), the use of systemic antibiotics within 2 weeks and systemic immunosuppressive therapy or phototherapy or systemic biologic agents within the previous 12 weeks prior to screening. Before skin sampling, the biopsy sites were left untreated for at least 2 weeks and cleansing with only the non-antibacterial Dove soap was allowed and washing was avoided for 24 h prior to sampling. The patients or healthy volunteers who did not match these clinical exclusion criteria were removed from the study. The following biological samples were then obtained and submitted to analysis: (1) microbiome samples from upper/lower back, posterior thigh or buttocks (PSO, AD, HV) with no prior cleaning or preparation of the skin surface using sterile gloves to prevent cross-contamination were obtained placing a sterile ring (2.5 cm diameter) onto the appropriate skin area, 1.5 ml phosphate-buffered saline (PBS) was supplemented into the ring and the area sampled scraping a glass rod in a circular motion ten times to the left and to the right. Subsequently, the microbiome-enriched PBS was harvested and stored. In addition, mock samples containing only PBS were collected at each sampling time in order to assess contamination. (2) 6 mm punch biopsies from skin at the "microbiome" sites were taken in local anesthesia. Subsequently, samples were stored in RNAlater (Sigma-Aldrich) and

subjected to further analyses (Supplementary Fig. S1a). The study was approved by the appropriate local Institutional Review Boards (University of Helsinki, Dnro 91/13/03/00/2011; Heinrich Heine University Düsseldorf, 3647/2011; King's College London, 11/H0802/6) and all subjects provided written informed consent before participation.

Upper back and thigh posterior were chosen to represent skin areas with AD lesions, while upper back and lower back were chosen for PSO lesions, at the same time taking care in minimizing harm due to the surgical procedure of obtaining skin biopsies (Supplementary Fig. 1a). Only samples from active skin disease, i.e. lesional skin, were included in the study, and normal skin from HV was used as a control. In HV, samples were taken from sites corresponding to the sampling sites in AD and PSO. Identical SOPs were used in all three clinical centers and quality and homogeneity was assessed and confirmed in a pilot study prior to starting the major sample collection. Final analysis included 82 AD, 119 PSO and 115 HV samples integrating the microbiome with the transcriptome.

**16S rRNA gene sequencing and analyses.** DNA was extracted from the clinical swab and mock samples using Qiagen's Pathogen Lysis Tubes and the QIAamp UCP Pathogen Mini Kit (Cat.No: 19092) according to the manufacturer's instructions. In brief, sample pellets were resuspended in 500 µl Buffer ATL and vortexed for 10 min at maximum speed using Pathogen Lysis Tubes containing glass beads. The samples were transferred to fresh Beckman tubes and incubated in 16.5 mg/ml lysozyme (Sigma) for 30 min at 37 °C. Fifty microliters proteinase K was added and the samples were then incubated for 10 min at 56 °C. Addition of 250 µl of Buffer APL2 was followed by incubation at 70 °C for 10 min. Ten microliters RNA-grade glycogen (20 mg/ml, Thermo Scientific) were added to maximize DNA recovery. Ethanol was added to a final concentration of 25%. DNA was extracted and washed using spin columns, and subsequently eluted in 50 µl of Buffer AVE.

For 16S rRNA gene amplification and preparation for sequencing, 2.5 µl template were amplified in RT-PCR GradeWater (Life technologies), 3% DMSO, with 1× PCR HF buffer using Phusion Hot start II DNA polymerase, 200 µM dNTPs (all Thermo Scientific), and 500 nM custom primers (Eurofins MWG Operon). One universal forward primer (341f 5′-CCTACGGGNGGCWGCAG with adaptor B, Lib-L) was paired with one of 104 barcoded reverse primers (805r 5′-GACTACHVGGGTATCTAATCC with adaptor A, Lib-L) (Supplementary Table 5). Each barcode consisted of seven nucleotides, contained no homopolymers, and a pair of barcodes differed in at least two positions. Each PCR was run in triplicates and the PCR products from each sample were pooled. A negative control PCR reaction lacking template was included for all primer pairs in each run. The PCR was run for 30 cycles. The PCR products were purified from the reaction using Dynabeads® MyOne™ Carboxylic Acid (Life Technologies, Cat.No: 35401) and TruSeq precipitation buffer (16% PEG-6000, 1.5 M NaCl) on the Magnatrix 1200 (LBH Advanced Bioservices AB, Sweden). The purity of the amplicons was visualized on the Agilent 2100 BioAnalyzer using High sensitivity DNA chips and reagents (Agilent Technologies, Cat.No: 5067-4626) according to the manufacturer's instructions. DNA concentrations were measured by real-time PCR (KAPA Library Quantification Kits For Roche 454 GS Titanium platform, Cat. No: KK4821 and BioRad CFX96 Touch™ Real-Time PCR Detection System; C1000 Thermal cycler) according to the manufacturer's instructions with samples diluted 1:500, 1:1000, and 1:2000 in 10 mM Tris-HCl, pH 8.0. Extension time was 90 s. Finally, the samples were adjusted to $1.0 \times 10^8$ DNA molecules for each sample before pooling 50–60 samples per 454 sequencing run.

For 454 amplicon sequencing, emulsionPCR was performed on the amplicon library using a large volume emPCR (Lib-L, v2 reagent kit) according to the manufacturer's amplicon protocols and pyrosequenced (one way read direction) on a Genome Sequencer FLX-Titanium instrument (Roche/454 Life Sciences) at Science For Life Laboratory (SciLifeLab), Stockholm. Each library was sequenced in both regions of a two region gasketed 70 × 75 mm Titanium PicoTiterPlate, and base calling was performed with the on-instrument amplicon filter settings. Samples containing only water were sequenced in order to assess contamination during the sequencing process.

For demultiplexing and preprocessing of 454 reads, all sequence reads were assigned to their samples using the unique sample barcodes. Raw sequence reads were analyzed with AmpliconNoise version 1.25 to remove 454 sequencing and PCR artifacts and PerseusD from the same program package to remove PCR chimaeras, using default parameter values. The output from each sample was further processed in QIIME (Quantitative Insights Into Microbial Ecology) version 1.8.0 if the number of processed high-quality reads exceeded 3000 per sample. Otherwise, the sample was resequenced.

For OTU clustering and taxonomy assignment, the preprocessed dataset comprised of a total of 3,357,091 high-quality reads. The following analysis steps were performed using QIIME version 1.8.0. OTUs were picked at 99.3% identity using the pick_open_reference_otus.py command and uclust 1.2.22q. Taxonomy was assigned using blast-2.2.22. The reference data files used for both OTU clustering and taxonomy assignment were downloaded from the Greengenes Database Consortium[55]. As AmpliconNoise did not perform very well in identifying chimeric sequences in our dataset, ChimeraSlayer[56] was applied here within the QIIME pipeline and identified chimeric sequences were removed from the OTU table and the phylogenetic tree.

**Analysis of 16S rRNA gene data.** The protocol for 16S rRNA gene data production used long amplicons and deep 454 sequencing in order to make it possible to exhaustively distinguish as many different OTUs as possible and to discover close to all relevant bacteria in the skin. The noninvasive sampling protocol was designed to specifically sample bacteria on the surface of the skin as the total amount of bacteria in the deeper skin layers is expected to be low. 16S rRNA gene PCR was carried out directly from the DNA preparation, without another amplification step to avoid amplification-induced bias to the data. The analysis of the 16S rRNA gene data was carried out according to standard protocols and relevant controls and low-quality data was removed using AmpliconNoise, which is the gold standard in the field. The same, easily reproduced sampling protocol was used in all cases and the DNA preparations, PCRs, sequencing and analysis were carried out in one facility using standard methods.

Three samples of poor quality were removed from the OTU table (1 PSO and 2 HV, Supplementary Fig. 1b). Abundances were normalized using the Trimmed Mean of M-values method (TMM), implemented in the edgeR Bioconductor package. Good's coverage and Shannon diversity index was calculated using QIIME version 1.8.0.

To preserve statistical power, only OTUs present in more than 25% of all samples were analyzed. Differentially abundant OTUs were identified from the filtered set by comparing the abundance distribution of each OTU across the three clinical groups (HV vs. PSO vs. AD) with the Kruskal−Wallis test (FDR, $p < 0.05$). Microbe–disease-specific associations were detected testing for differences in the abundance distribution of each differentially abundant OTU between the diseased (PSO and AD) and HV groups with the Mann−Whitney $U$ test (FDR, $p < 0.05$).

For analysis of confounding variables, associations between OTU abundance and nonclinical factors (age, gender, body site and institution) were tested for within the HV group using: (1) Kruskal−Wallis test for body site and institution; (2) Mann−Whitney $U$ test for gender and (3) Spearman correlation for age. FDR correction was used to correct for multiple testing. Additionally, the same association tests were carried out within the PSO and AD groups to search for disease-factor interactions. Then, a linear model was fitted for each OTU to account for the discovered nonclinical factor associations in the microbe−disease association test. The general linear model equation is: OTU_abundance = clinical_group + nonclinical factors and it was fitted using the lm function of R. $p$ values from the test for the disease were FDR-corrected for multiple hypothesis testing.

Bacterial aerobes vs. strict anaerobes were determined according to Bergey's Manual of Systematic Bacteriology[57]. Anaerobe abundance comparison between AD and HV was performed using a Mann−Whitney $U$ test. Additionally, to account for the systematic increase of *S. aureus* in AD, the same test was repeated only with samples in which the aforementioned bacterium was not detected.

For visualization of results, the phylogenetic tree in Fig. 1a was calculated using ClustalW2 and it was visualized, along with associated data using interactive Tree of Life, iTOL[58] and Adobe Illustrator. NMDS analysis was performed using phyloseq. Boxplots were generated in R version 3.1.0 with the ggplot2 package version 1.0.1, with whiskers extending to max 1.5 IQR. Outliers were removed.

**Analysis of 16S rRNA gene sequencing negative controls.** For OTU clustering and taxonomy assignment, OTU picking was performed independently from the volunteer samples. The pick_closed_reference_otus.py pipeline from QIIME version 1.9.1 was run on the samples, with the same parameters and the same version of the Greengenes 16S database used to process the volunteer samples. The procedure yielded a total of 425 identified OTUs in 40 samples.

To reduce false positives in the contamination analysis, OTUs were discarded if they were present in less than five samples or if the raw abundance of the OTU in any sample was less than 100 reads. Filtering criteria were selected to preserve both low abundant bacteria whose presence is supported by many negative controls and bacteria with enough read coverage to be trustworthy. The filtering procedure reduced the total number of detected OTUs in the mock samples from 425 to 73.

OTU ids obtained from the negative controls were compared against the list of differentially abundant OTU ids from the volunteer samples. None of the OTUs from the negative controls matched the OTUs identified in the volunteer samples.

**Generation and analysis of metagenomics shotgun data.** Libraries for shotgun sequencing were prepared from the same DNA preparations that were used for 16S rRNA gene PCR. The libraries were prepared using the ThruPLEX DNA-seq kit (Rubicon Genomics, Cat No. R400406) and sequenced using the Illumina HiSeq technology. The resulting reads were quality controlled and human sequences were removed. Primary analysis was carried out using MetaPhlan2 and humann2.

**16S rRNA gene sequence-based metagenomic feature inference.** Metagenome inference from 16S rRNA gene sequences was carried out using PICRUSt[59]. Briefly, gene content of individual OTUs was inferred using the Greengenes v.13.5 database and KEGG Orthologs (KOs) were extracted on multiple hierarchical levels (KEGG tiers 1–4). Relevant microbial pathways were extracted from the tier 1 categories "metabolism" and "signaling and cellular processes". Contribution of each OTU to the respective functional terms was quantified using the "metagenome_contributions.py" subroutine, where the relative contribution of OTUs was compared

between *S. aureus* high and *S. aureus* low groups using Student's *t* test followed by Benjamini−Hochberg multiple testing correction. Visualization was carried out using STAMP software[60] and R v.3.5.1.

**Generation of microarray transcriptional profiles**. The skin biopsy samples were stored in RNAlater and total RNA was extracted from the tissue samples using the RNeasy Fibrous Tissue Mini kit (Qiagen). Tissue samples were homogenized using the FastPrep-24 instrument (Nordic Biolabs AB), and RNA was extracted according to the manufacturer's instructions. The yield and purity of RNA in the samples were controlled using a Nanodrop spectrophotometer and Qubit fluorometer to verify absence of inhibitors (R260/280: 2.1; R260/230 nm: 1.3). RNA integrity was quantified by electrophoresis and performed using Agilent dedicated Lab-on-chip (RNA6000 Nano and Pico kits). RNA Integrity numbers and 28S/18S ratio averages were respectively 8.6 and 2.

One hundred nanograms of total RNA was amplified according to Affymetrix protocols (Affymetrix® GeneChip® Whole Transcript (WT) Expression Arrays). Based on expertise of Institut Curie genomic platform, MAQC A RNA samples (Universal RNA, Stratagene, P/N: 740000) were implemented to series of RNA amplification in order to monitor target preparation. In practice, series of 47 RNA (from healthy volunteers, and patients) and 1 Universal RNA were amplified, monitored and labeled. During synthesis steps, purified molecules were quantified using a multichannel Nanodrop (ND8000, Thermo) to normalize amount of molecules used for DNA synthesis (10 μg) and hybridization (5.5 μg). Molecules were also controlled on high throughput electrophoresis (QIAxcel DNA, Qiagen) in order to monitor the size of complementary RNAs (average: 500nt), and fragmented DNA (average: 50nt), to ensure quality of targets and hybridization of microarrays. Series of 96 targets were hybridized onto Affymetrix Gene ST 2.1 96 array plates, including in total two Universal RNA, using an Affymetrix Genetitan MC system. Quality of raw data and normalized data was monitored to control dynamics of the measurements, across series of synthesis, and series of hybridization using bacterial spike in controls added to total RNA, and using Universal RNA.

An automated quality control pipeline based on the arrayQualityMetrics method was used to capture quality failures in microarray data. A total of 12 samples were removed from the dataset. Data were then normalized using the Robust Multi-array Average (RMA) approach implemented in the affy Bioconductor package[61].

Technical batch effects originating from sample preparation for microarray analysis were removed (SVA-package with Combat function). For identification of differentially expressed genes, a linear model (R package *Limma*) was fitted to the data (using age, gender, anatomical location and clinical center as covariates), and pairwise comparisons were done using the empirical Bayes method. Transcriptomes were defined based on a fold change of 1.5 or greater and a Benjamini−Hochberg adjusted *p* value less than 0.05. Functional enrichment analyses were performed using web-based tools (http://amp.pharm.mssm.edu/Enrichr/ and the PANTHER Classification System version 13.1) and the Ingenuity Pathway Analysis (IPA, QIAGEN Redwood City, www.qiagen.com/ingenuity).

High RNA and microarray quality was assured by standard methods including measurement of RNA quantity and degradation, the use of control probes and QC and preprocessing methods provided the Bioconductor repository of R libraries. The high quality of samples is reflected by a very low rate of exclusion due QC on S16 (3/331 subjects) or Affymetrix (12/331 subjects) analyses. Overall the data were robust and reliable.

**3D human epidermal equivalents and bacterial stimulations**. Human epidermal equivalents (HEEs) were generated by seeding 300,000 human primary keratinocytes (obtained from abdominal plastic skin surgery) on polycarbonate cell culture inserts, followed by 11 days of culture (3 days submerged and 8 days air-exposed) (Supplementary Fig. 13). The last day of culture 10^6 CFU *S. aureus* was topically applied on the HEEs for 24 h, followed by analysis of morphology, histology (protein expression) and qPCR analysis (gene expression).

For bacterial cultures, *S. aureus* (ATCC 29213) was obtained from the Department of Medical Microbiology of the Radboudumc. Bacteria were inoculated on Columbia agar with 5% sheep blood (Dickinson and Company (BD), Sparks, MD) overnight (*S. aureus*) or for 4 days (*F. magna*) at 37 °C. One single colony of each plate was picked and cultured in Brain Heart Infusion medium (Mediaproducts BV, Groningen, The Netherlands). Bacteria were collected by centrifugation, washed two times with PBS and finally resuspended in PBS resulting in bacterial concentrations of 10^7 CFU/ml. To determine the amount of bacteria that was brought on the keratinocyte cultures, bacterial suspensions were serially diluted in steps of 5. Ten microliters of each dilution was placed on sheep blood agar plates and incubated overnight or for 4 days at 37 °C in aerobic (*S. aureus*) conditions, respectively. Visible colonies on the plate were counted for each dilution. The number of CFU was calculated: counted CFU × dilution factor.

For the development of 3D HEE, primary human keratinocytes were obtained from abdominal plastic skin surgery, and isolated and expanded according to the Rheinwald−Green protocol[62] and stored in liquid nitrogen. For the generation of HEE's 300,000 primary human keratinocytes were seeded onto 0.4 μm pore size transwell filters (Thincerts, Greiner Bio-One). Prior to seeding, these transwells were incubated with 100 μg/ml rat-tail collagen (BD Biosciences) diluted in sterile

cold PBS at 4 °C for 1 h. Afterwards, excessive coating was carefully aspirated and filters were washed once with cold, sterile PBS. First, the primary keratinocytes were cultured in a submerged manner in proliferation medium (CnT-prime, CELLnTEC) for 2 days to form a homogenously distributed monolayer. One day later, medium was changed to differentiation medium (60% 3D barrier, CELLnTEC complemented with 40% Dulbecco's modified Eagle medium, Sigma). On the third day the transwell inserts were lifted to the air−liquid interface to induce differentiation and stratification of the epidermis. From now on cells were cultured in differentiation medium until the end of the culture and they were refreshed with differentiation medium every other day.

On day 7 of air−liquid interface culture 10^6 CFU *S. aureus* in PBS (cultured as described above) was applied topically on the HEEs. For control HEEs only PBS was applied. After 24 h HEEs were harvested for qPCR analysis.

RNA isolation, cDNA synthesis and qPCR analysis was performed as described earlier[63]. All primers were designed and used as described previously[64] (Supplementary Table 5). Target gene expression was normalized to the expression of the house-keeping gene human acidic ribosomal phosphoprotein P0 (*RPLP0*). The ΔΔCt method was used to calculate relative mRNA expression levels[65].

For morphological analysis, HEEs were fixed in 4% formalin solution for 4 h and embedded in paraffin. 6 μm Six-micrometer sections were stained with hematoxylin and eosin (H&E, Sigma-Aldrich) or DAPI stain solution (DAPI Fluoromount-G, Southern Biotech) as described previously[66].

**Tryptophan dependence assay**. The growth of *S. aureus* strains isolated from moderate-to-severe atopic dermatitis patients was analyzed in tryptophan-containing vs. tryptophan-depleted culture media. The tryptophan-depleted, conditioned medium was derived from IFN-γ activated, IDO-positive human glioblastoma cells 86HG39 that efficiently inhibit the growth of tryptophan-auxotrophic *S. aureus* strains[67]. 86HG39 cells were cultured in Iscove's modified Dulbecco's medium (IMDM) (Gibco, Grand Island, USA), supplemented with 5% heat-inactivated fetal calf serum (FCS) in culture flasks (Costar, Cambridge, USA) in a humidified incubator (37 °C, 10% CO₂). 1 × 10^6 cells per cell culture flask were stimulated with 1000U/ml IFN-γ (R&D Systems, Minneapolis, USA) for 74 h and the conditioned medium was harvested. As controls, medium from unstimulated 86HG39 cells or tryptophan-free RPMI 1640 Medium (Gibco, Grand Island, USA) (with or without tryptophan supplementation) were used. For the tryptophan dependence assay, a 24 h old colony was picked, resuspended in PBS (Gibco, Grand Island, USA) and serial diluted. The conditioned medium as well as control media (200 μl) were inoculated with 10 μl of the bacterial dilution containing 10–100 colony forming units (cfu). Cultures were incubated overnight. Bacterial growth was monitored by measuring the optical density of resuspended cultures at 620 nm. Tryptophan dependency was validated by growth restoration in tryptophan-containing cultures.

**Classification and feature selection**. In order to identify the best set of representative microbiota for disease class, we used a supervised learning approach based upon Random Forest classification models. The pipeline was used to train models for three contrasts (Healthy vs. AD, Healthy vs. PSO and PSO vs. AD) with the aim of identifying the features that best discriminate disease groups. To identify the most stable predictors of disease, we first removed under-represented species by filtering out those that were present in less than 15 samples for each contrast. Next, to select the most discriminant OTU features, Random Forest feature selection implemented in the R package Boruta was used under a tenfold cross-validation framework with ten randomized repeats. Selected features were ranked by Boruta according to the variable importance score Z, which represents the average loss in accuracy after permutation of attribute values across samples. OTUs with a mean Z score greater than 0.2 were considered for further analysis. The fold change between healthy and disease patients was calculated to identify depletion or increased abundance in disease. After selection of features, the classification model was built within the R package randomForest. We evaluated the performance of the model using the R package ROCR and report the mean AUC across all folds[68]. NMDS with Bray−Curtis distance using the selected feature sets with the function metaMDS from the R package Vegan was used for visualization purposes only. Selection frequency was calculated as the percentage of times a variable was selected by Boruta across all folds over all randomizations.

**Microbial co-occurrence networks**. We constructed interaction networks on all OTUs present in at least 5% of samples resulting in a core microbiome of 569 OTUs. We employed SparCC[19] on the raw OTU abundance, which calculates a corrected correlation coefficient designed specifically for assessing the correlative relationships between taxa in microbiome studies. The statistical significance of correlations was evaluated against an empirical null distribution obtained with 100 bootstrap iterations (*p* < 0.05 and SparCC > 0.2). Network visualizations were generated in Cytoscape.

**Dimensionality reduction of transcriptomics data**. To select genes for integration, differential analysis between healthy and diseased samples was performed using the limma package reducing the total number of host transcripts under consideration from 32,633 to 16,716 AD and 22,433 PSO-associated transcripts

(FDR < 0.05). Co-expression networks were constructed using the pairwise Pearson correlations between differentially expressed genes. Edges were determined by a hard threshold of $r < 0.7$ and the main component of the network was retained for further analysis. This reduced the overall transcriptome to a core disease-associated network of 1833 AD and 2653 PSO-associated genes. To partition the network into modules of genes that display similar expression profile, the Louvain method for community detection was employed. Co-expression networks were visualized in Cytoscape.

**Identification of host−microbial interactions**. For each module identified through network community detection, feature reduction was performed by calculation of the module eigengene which is equivalent to the first principal component of module gene expression[69]. To test for associations between microbial taxa and module eigengenes, we used the MaAsLin (https://huttenhower.sph.harvard.edu/maaslin) package[21] which fits linear models between arcsine square root transformed OTU relative abundances and metadata after removal of statistical outliers. We assessed the relationship between the top 25 most significant disease-associated microbes and module eigengene pairs while correcting for potential confounding factors using the model: microbe ~ module + anatomical location + institution + age + gender. $p$ values were corrected using the Benjamini−Hochberg method and microbe−module associations with FDR < 0.20 were considered as significant. In AD, the top most abundant microbe (*S. aureus*) was used for stratification of the patient cohort, and differential gene expression was compared between samples exhibiting the top and bottom tertiles of *S. aureus* abundance. Network modules enriched for *S.aureus*-associated genes were identified using hypergeometric tests followed by Benjamini−Hochberg multiple testing correction. For modules with significant microbial association, we evaluated the overrepresentation of pathways using Ingenuity pathway analysis.

**Microbiome, transcriptome and disease severity associations**. Regression algorithms were employed for the task of identifying gene and taxonomical markers that can predict the severity of disease indices. For transcriptomic data, one score is computed for each gene, quantifying its degree of correlation with severity. The genes are then ranked in descending order according to their scores and a series of regression models are trained onto the top $N$ genes ($N = 5$−50 *in increments of five*). The trained model can then be used to predict the severity of new samples. To ensure that the constructed regression model can generalize well to unseen samples, a leave-one-out cross-validation scheme is applied to identify performance. Briefly, one sample is made blind to the process of calculation of correlation and training of regression model, and is used to estimate the accuracy of the trained model. In leave-one-out cross-validation, each sample is held out exactly once and the prediction errors are averaged over all samples. The optimal number of genes is then chosen as the set that corresponds to the lowest mean absolute prediction error. The same procedure was applied to OTU abundance data to identify a signature that correlates with severity of disease. Six multivariate regression methods were applied including linear regression, SVM, 5-NN, M5P, RF and Cubist and the model with the lowest mean absolute error was chosen.

**Reporting summary**. Further information on research design is available in the Nature Research Reporting Summary linked to this article.

## Data availability

Data underlying Figs. 1−4, Supplementary Figs. 1−12, 14−16 and Supplementary Tables 1−3 are available from the NCBI Sequence Read Archive under accession PRJNA554499 and from EBI Array Express under accession E-MTAB-8149. Data underlying Supplementary Fig. 13 is provided as a Source Data file. All other data are available from the corresponding author upon reasonable requests.

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

## Acknowledgements

The authors want to thank Prof. A. Andersson (KTH Royal Institute of Technology and Science for Life Laboratory (SciLifeLab)) for providing the 16S primer designs and acknowledge support from SciLifeLab, the national infrastructure SNISS, and Uppmax for assistance in massive parallel sequencing and computational infrastructure. Technical support by D. Lundin, BILS (Bioinformatics Infrastructure for Life Sciences) and T. Keyvanfar is also acknowledged. The research has received funding from the FP7/2007–2013 (Grant 261366). The study was partially funded by the Knut and Alice Wallenberg Foundation, the Department of Health via the National Institute for Health Research (NIHR) comprehensive Biomedical Research Centre award to Guy's & St Thomas' NHS Foundation Trust in partnership with King's College London and King's College Hospital NHS Foundation Trust (guysbrc-2012-1) Trust, and Dunhill Medical Trust, the Association pour la Recherche contre le Cancer (ARC), the European Research Council (Grant IT-DC 281987), Institute National de la Santé et de la Recherche Médicale (BIO2012-02 and BIO2014-08), INCA (2011-1-PL BIO-12-IC-1), Fondation ARSEP (R12023JJ), ANR (ANR-13-BSV1-0024-02, ANR-10-IDEX-0001-02 PSL*, ANR-11-LABX-0043 and ZonMw MKMD grant 114021503) and BIOMAP IMI2 (Grant 821511). Open access funding provided by Karolinska Institute.

## Author contributions

Conceptualization, H.A., B.A., J.B., B.H., J.K., A.L., F.L-S., F.N. A.R., J-M.S., V.S., S.T.; Resources, J.B, B.H., A.L., A.R., A.M.B., M.F., K.J., E.L., S.M., S.S., I.T.; Methodology, J.K., T.S., S.P.-N., B.A.; Investigation, S.P.-N., H.D., T.S.; Formal analysis, M.B., M.J., S.P.-N., V.S., S.T., M.B-S., M.J., G.M, L.Y., D.G., N.F., H.A., D.P., P.O., E.v.d.B, P.Z., G.R., J.S., T.E., W.D., S.K., H.S., H.N.; Data curation, P.H., G.J-C., V.S.; Visualization, B.A., H.A., N.F., M.J., G.M., S.P-N., S.T., F.N., V.S.; Supervision, B.A., F.N., S.T., V.S., J-M.S.; Writing—original draft, H.A., B.A., N.F., B.H., M.J., A.L., G.M., F.N., V.S., S.T.; Writing—review and editing, H.A., B.A., B.H., N.F., G.M., S.T., M.F., R.S.G., P.K., S.K., J.K., A.L., F.L-S., D.P., S.P-N., J-M.S., U.G., B.H., A.R., H.S.; Equal contribution: S.T., A.L., V.S., F.N., B.H., B.A., H.A.; Project administration, H.A., N.F.

## Additional information

**Competing interests:** The authors declare no competing interests.

**Peer Review Information** *Nature Communications* thanks the anonymous reviewers for their contribution to the peer review of this work. Peer reviewer reports are available.

Nanna Fyhrquist [1,2,26], Gareth Muirhead[3,4,26], Stefanie Prast-Nielsen [5,26], Marine Jeanmougin[6,7,8,9,26], Peter Olah[10,11], Tiina Skoog [12], Gerome Jules-Clement[6,7,8,9], Micha Feld[10], Mauricio Barrientos-Somarribas[13], Hanna Sinkko[1,2], Ellen H. van den Bogaard[14], Patrick L.J.M. Zeeuwen[14], Gijs Rikken[14], Joost Schalkwijk[14], Hanna Niehues [14], Walter Däubener [15], Silvia Kathrin Eller[15], Helen Alexander[16], Davide Pennino[4], Sari Suomela[17], Ioannis Tessas[17], Emilia Lybeck[17], Anna M. Baran[10], Hamid Darban[13], Roopesh Singh Gangwar [18], Ulrich Gerstel[19], Katharina Jahn[10], Piia Karisola[2], Lee Yan[3], Britta Hansmann[19], Shintaro Katayama[12], Stephan Meller[10], Max Bylesjö[20], Philippe Hupé [6,7,8,21], Francesca Levi-Schaffer[18], Dario Greco [22,23,24], Annamari Ranki[17], Jens M. Schröder [19], Jonathan Barker[16], Juha Kere [12,25], Sophia Tsoka[3], Antti Lauerma [17], Vassili Soumelis [6,9], Frank O. Nestle[4], Bernhard Homey[10], Björn Andersson [13] & Harri Alenius [1,2]

[1]Institute of Environmental Medicine, Karolinska Institutet, Stockholm 17177, Sweden. [2]Department of Bacteriology and Immunology, Medicum, University of Helsinki, Helsinki 00014, Finland. [3]Department of Informatics, Faculty of Natural and Mathematical Sciences, King's College London, London WC2R 2LS, UK. [4]Cutaneous Medicine Unit, St. John's Institute of Dermatology and Biomedical Research Centre, Faculty of Life Sciences and Medicine, King's College London, London SE1 9RT, UK. [5]Centre for Translational Microbiome Research (CTMR), Department of Microbiology, Tumor and Cell Biology, Karolinska Institutet, Stockholm 17177, Sweden. [6]Institut Curie, 26 rue d'Ulm, 75248 Paris, France. [7]INSERM, U900, 75248 Paris, France. [8]Mines ParisTech, 77300 Fontainebleau, France. [9]INSERM, U932, 75248 Paris, France. [10]Department of Dermatology, University Hospital Duesseldorf, Duesseldorf 40225, Germany. [11]Department of Dermatology, Venereology and Oncodermatology, University of Pécs, Pécs 7632, Hungary. [12]Department of Biosciences and Nutrition, Karolinska Institutet, Stockholm 17177, Sweden. [13]Department of Cell and Molecular Biology, Science for Life Laboratory, Karolinska Institutet, Stockholm 17177, Sweden. [14]Department of Dermatology, Radboud University Medical Center, Radboud Institute for Molecular Life Sciences, Nijmegen 6525, The Netherlands. [15]Institute for Medical Microbiology and Hospital Hygiene, Heinrich Heine University Duesseldorf, Duesseldorf 40225, Germany. [16]St John's Institute of Dermatology, Division of Genetics and Molecular Medicine, Faculty of Life Sciences and Medicine, Kings College London, London SE1 9RT, UK. [17]Department of Dermatology, Allergology and Venereology, University of Helsinki and Helsinki University Hospital, Inflammation Centre, Helsinki 00250, Finland. [18]Pharmacology Unit, School of Pharmacy, The Institute for Drug Research, Faculty of Medicine, The Hebrew University of Jerusalem, Jerusalem 91120, Israel. [19]Department of Dermatology, University Hospital Schleswig-Holstein, Kiel 24105, Germany. [20]Fios Genomics, Edinburgh EH9 3JL, UK. [21]CNRS, UMR144, 75248 Paris, France. [22]Faculty of Medicine and Life Sciences, University of Tampere, Tampere 33520, Finland. [23]Institute of Biomedical Technology, University of Tampere, Tampere 33520, Finland. [24]Institute of Biotechnology, University of Helsinki, Helsinki 00014, Finland. [25]School of Basic and Medical Biosciences, King's College London, London SE1 9RT, UK. [26]These authors contributed equally: Nanna Fyhrquist, Gareth Muirhead, Stefanie Prast-Nielsen, Marine Jeanmougin.

