## [Peer Review File · Nature Communications]

Reviewers' comments:

Reviewer #1 (Remarks to the Author):

Fyhrquist et al. perform a matched 16S rRNA and host transcriptome profiling on AD, psoriasis, and healthy controls at two different skin sites of disease occurrence. Major findings cited confirmed previous reports that *Staphylococcus aureus* is significantly associated with AD (Kong et al) with depletion of anaerobes (Myles et al), and that a strong microbial signature is lacking in psoriasis. Reassuringly, the transcriptional data is largely consistent with previous reports of AD or psoriasis transcriptomes, and the authors found that the host transcriptome was a stronger predictor of skin disease severity than different microbiome metrics.

We commend the authors on the size of the cohort, the care taken with negative controls and confounders, and a clear and well-presented paper. However, the study itself represents a relatively modest advance for the field, with the major contribution being a correlation analysis with microbial features with host skin transcriptome data. The size of the cohort is an asset and allows the author to break down the AD patients into yes/no *S. aureus* groups which allows them to further refine transcriptome correlates. Like these previous studies, different classifier methods were used to define host or microbial features characteristic of the disease, with the major advantage again being the size of the study.

A major point of revision would be to include additional covariate analysis for different skin treatments that the patients had undergone prior to the washout period with Dove, and to describe with more detail the different features of the matched healthy controls. This is because the authors observed that there were likely different community states/microbiome 'types' for AD because some individuals were characterized by a prevalence of *S. aureus* while others not. Kong et al. reported that intermittent treatment shifts the microbiome towards one with significantly less *S. aureus*, and it is likely that this could be one of the factors that could explain the variation.

Also, why were OTUs clustered at the 99.3% identity level? Which is fairly atypical and naturally resulted in a very large number of OTUs. There should be some justification of this in the methods, and/or also performed at more conventional levels (e.g., 97%).

Reviewer #2 (Remarks to the Author):

In the paper by Fyhrquist et al., “Microbe-host interplay in atopic dermatitis and psoriasis”, the authors assemble a large cohort of cases and controls (over 300 individuals), for which they perform 16S rRNA sequencing and microarray-based analysis of transcriptomes. They find that AD is dominated by *S. aureus* and immune activation, whereas PSO was associated with assemblages of organisms.

I was able to review the technicality of the manuscript as my expertise is in microbial community analysis and gut microbiomes, though I was able to only touch upon the conclusions regarding changes in the immune system pertaining to skin disease and the originality of the work from someone in the microbiome field, but an outsider to these specific areas.

My biggest issue with the manuscript is that it overall lacked details about the study design, sample collection and most importantly, statistical analyses, which made it both hard to review and also gave the impression that many of these choices were made without sufficient consideration. These include the method of DNA extraction (crucial for comparing across studies), the methods for performing skin swabs (Fig. S1 implies some buffer but none was discussed), whether there were any exclusion criteria, how samples were matched, and what affymetrix plates were used for transcriptomics, to name a few. In terms of statistical analysis, it was unclear why some choices were made—99.3% identity for clustering, how controlling for confounders was done, but also what tests were used (see Figure 1A, a p-value is reported with a vague descriptor of ‘nonparametric score’, but the text only says ‘significantly associated’, yet this repeats in multiple results of the paper, and this is unclear whether this is significantly associated with one of the conditions or if multiple are being compared etc.). Similarly, in the transcriptomics section, the only thing that was mentioned was “Differential gene expression analysis” (line 347) rather than the specific method used. There wasn’t mention about false-discovery corrections for some analyses, though mentioned for others (i.e. Fig 4), or whether any normalization was performed on gene abundances. Since the devil is in the details for microbiome analyses, this can potentially be a major issue.

My other issue with the manuscript is the novelty of the results. Despite my disclaimer that I don’t work entirely in this field, it seems obvious both from quick lit searches and their own statements that there have been a number of studies that have done community analysis of psoriasis and AD and their analysis doesn’t particularly add much. The *S. aureus*-association with AD is not a new finding, and this has been reported on many times. There have also been RNA-seq experiments performed on both AD (Suarez-Farinas et al, 2015 and even a metaanalysis of transcriptome analyses (Ding et al., 2015) and PSO (Li et al., 2014). They find many of the same co-occurrences and pathway regulation, even according to their own statements (lines 552, 574). It was hard to decipher whether direct comparison could be made between the two disease cohorts because of the ambiguity of the matching scheme, and also the difference in body sampling sites. I can’t comment whether their results significantly improve or change our understand of PSO or AD.

Other issues:

- It was not clear how samples were matched, especially since large differences in gender underlie the Psoriasis and HV cohorts. Similarly, age-related dysbioses have been reported for skin microbiomes which are thought to predispose individuals to AD. This wasn't tested (or mentioned) within this cohort.
- Were there any exclusion criteria? Atopic dermatitis and psoriasis can be treated in many ways, including topical medications (including corticosteroids), oral immunosuppressant therapies, antibiotics, vitamin therapies, and oral steroids as well as immunomodulators. These were never mentioned in the patient recruitment or analysis.

Minor issues:

- Hard to follow what is being plotted in many of the figures. I mentioned Fig. 1A above, but this is the same for Fig 1B. Is this some aggregate of all of the samples, masking inter-personal variability? This is abnormal to group individual compositions into one composite composition.
- The data display was very confusing—like Figure 3B is very confusing—are these genes up or down in these different conditions and the same direction in these different conditions. This was a confusing figure. Figure 3C/D are also confusing visually. Or what are the yellow edges in Fig 4F?

Reviewer #3 (Remarks to the Author):

In this manuscript, the authors conducted 16s rRNA sequencing identification of the skin microbiome of patients with atopic dermatitis and psoriasis as well as host transcriptome characterization using microarrays of skin biopsies at the same sites. They showed that *S. aureus* is significantly associated with atopic dermatitis and that multiple species, such as *Corynebacterium* spp and anaerobes are associated with psoriatic skin. They also made classifiers to identify the microbial species that could most differentiate between atopic dermatitis vs normal and psoriasis vs normal skin. They then analyze the host skin transcriptome at these sites.

Many of the findings in this paper confirm previously known studies. The skin microbiome findings for atopic dermatitis reflect what is already known in the literature^{1,2}. Some findings for the psoriatic skin microbiome are also already known, such as the overrepresentation of *Streptococcus* spp and overabundance of *Corynebacterium* species (reviewed in 3) although this study has a

different way of presenting the data (Figures 1-2). The approach of defining microbial classifiers of AD and psoriatic skin is novel, but without any mechanistic data to suggest why particular species in the classifier are disease-defining, the finding has less impact. Additionally, even if species within the classifier are not disease-defining but are in fact secondary effects of a particular type of inflammation, without mechanistic data suggesting why AD vs psoriatic inflammation might bias towards one set of bacteria vs another, it is hard to assess the biological significance of the classifier.

The atopic dermatitis transcriptomic data reveal similar pathways as previous skin transcriptome studies^{4,5} and confirms known involved pathways, such as Th1 and Th2 pathways (Figure 3). The psoriasis transcriptomic data also confirm known involved pathways and prior studies (reviewed in 6,7), such as IL-17 signaling and IFN- γ signaling (Figure 3). Although Figures 3-4 were clear and beautiful representations of the data, they did not offer new hypothesis-generating insight into disease pathogenesis.

The main novelty of this study is that skin microbiome and transcriptome data were collected at the same time, to allow for analysis of microbiome-transcriptome correlations. However, in their analysis, the authors did not reveal surprising information. By comparing the transcriptomes in AD samples with high abundance of *S. aureus* to those with low abundance of *S. aureus*, the authors found an enrichment of pathways such as "Keratinization" and enrichment in gene expression of IL18, IL1a, TNF, IFN γ , and other expected pro-inflammatory cytokines (Figure 4C-D).

The analysis of psoriatic microbiomes to transcriptomes revealed that *Corynebacterium* spp may be negatively associated with inflammatory pathways in psoriasis (p.17, line 441-461). However, there are no mechanistic studies to suggest that *Corynebacterium* is protective. Additionally, earlier in the microbiome study, *Corynebacterium* was shown to be overrepresented in psoriatic lesions, which seems contradictory. Perhaps, some *Corynebacterium* species are immunoregulatory or help resistance towards more inflammatory bacteria and are therefore protective, whereas other *Corynebacterium* species are positive associated with psoriasis because they are more pro-inflammatory. This idea would be interesting, but would need additional studies in order to be validated.

There is another finding in the paper that I thought was interesting: abundance of anaerobic bacteria in AD was decreased suggesting that there is increased O₂ tension in AD lesions (Figure 1D, discussed in line 555-557). An analysis of metabolic pathways upregulated in disease states was a very successful approach to show that utilization of formate and aerobic respiration by *E. coli* was increased in gut inflammation⁽⁸⁾ and then that blocking this particular microbial metabolic pathway using tungsten could ameliorate colitis⁽⁹⁾. Therefore, if the authors were to use metagenomic sequencing instead of 16s rRNA sequencing to look at the microbial metabolic pathways that were changed in disease states and correlate this to host transcriptome signatures, this study would be much more informative and more likely to generate hypotheses for further mechanistic studies.

The approach and intent of the study was commendable and as a field, we do need to move toward connecting microbiota changes to host epithelial or immunologic changes. However, the data presented here did not reveal many distinctive positive results. If the authors were to re-focus on some of the more interesting findings highlighted above and develop further mechanistic insight instead of presenting a broad overview of findings without specific hypotheses, this would greatly improve the impact of this study.

References:

1. Byrd, A. L. et al. *Staphylococcus aureus* and *Staphylococcus epidermidis* strain diversity underlying pediatric atopic dermatitis. *Sci. Transl. Med.* 9, eaal4651 (2017).
2. Kong, H. H. et al. Temporal shifts in the skin microbiome associated with disease flares and treatment in children with atopic dermatitis. *Genome Res.* 22, 850–859 (2012).
3. Yan, D. et al. The Role of the Skin and Gut Microbiome in Psoriatic Disease. *Curr. Dermatol. Rep.* 6, 94–103 (2017).
4. Ghosh, D. et al. Multiple Transcriptome Data Analysis Reveals Biologically Relevant Atopic Dermatitis Signature Genes and Pathways. *PLOS ONE* 10, e0144316 (2015).
5. Ewald, D. A. et al. Meta-analysis derived atopic dermatitis (MADAD) transcriptome defines a robust AD signature highlighting the involvement of atherosclerosis and lipid metabolism pathways. *BMC Med. Genomics* 8, (2015).
6. Jiang, S., Hinchliffe, T. E. & Wu, T. Biomarkers of An Autoimmune Skin Disease—Psoriasis. *Genomics Proteomics Bioinformatics* 13, 224–233 (2015).
7. Lowes, M. A., Suárez-Fariñas, M. & Krueger, J. G. Immunology of Psoriasis. *Annu. Rev. Immunol.* 32, 227–255 (2014).
8. Hughes, E. R. et al. Microbial Respiration and Formate Oxidation as Metabolic Signatures of Inflammation-Associated Dysbiosis. *Cell Host Microbe* 21, 208–219 (2017).
9. Zhu, W. et al. Precision editing of the gut microbiota ameliorates colitis. *Nature* 553, 208–211 (2018).

Reviewer #1 (Remarks to the Author):

Comment 1: *Fyhrquist et al. perform a matched 16S rRNA and host transcriptome profiling on AD, psoriasis, and healthy controls at two different skin sites of disease occurrence. Major findings cited confirmed previous reports that Staphylococcus aureus is significantly associated with AD (Kong et al) with depletion of anaerobes (Myles et al), and that a strong microbial signature is lacking in psoriasis. Reassuringly, the transcriptional data is largely consistent with previous reports of AD or psoriasis transcriptomes, and the authors found that the host transcriptome was a stronger predictor of skin disease severity than different microbiome metrics. We commend the authors on the size of the cohort, the care taken with negative controls and confounders, and a clear and well-presented paper. However, the study itself represents a relatively modest advance for the field, with the major contribution being a correlation analysis with microbial features with host skin transcriptome data. The size of the cohort is an asset and allows the author to break down the AD patients into yes/no S. aureus groups which allows them to further refine transcriptome correlates. Like these previous studies, different classifier methods were used to define host or microbial features characteristic of the disease, with the major advantage again being the size of the study.*

ANSWER: Thank you for reviewing our manuscript and for raising this comment. We would like to point out that next to its size a main novelty of this study is that the cutaneous microbiome and the host's transcriptome data were collected at the same location and time, to allow for analysis of microbe-host-interactions (microbiome-transcriptome correlations). The approach and intent of the study is commendable and as a field, we do need to move towards connecting changes in microbiota to host epithelial or immunologic homeostasis and disease.

We have now revised the manuscript and added/ included:

1. Preliminary data from whole genome sequencing (WGS), to validate 16S based OTU classification and functional predictions based on the 16S marker. The results show close agreement between the two sequencing methods (Fig. 1) and are now included in the manuscript in Fig.S9 and in the text on p.17.

Fig. 1. Validation of 16S-based OTU classification by WGS. Correlation of relative abundance of *S. aureus* between 16S rRNA sequencing and WGS metagenomics. For validation of 16S OTU classification, preliminary data of WGS metagenomic sequencing of 20 randomly selected samples from 10 *S. aureus* 'high' and 10 *S. aureus* 'low' patients were used. The significant correlation ($r^2=0.919$) shows agreement of taxonomic classification between the independent sequencing methods.

- Mechanistic validation of the observed host-microbe-interactions, including exposure of human 3D epidermal equivalents with viable *S. aureus*, showing the induction of key *S. aureus* signature genes (Fig. 2). The results are now included in the manuscript in Fig S10 and in the text on p. 17.

Fig. 2. Mechanistic validation of host-microbe interactions. **a** Morphological analysis of human epidermal equivalents (HHEs). **b** H&E and **c** immuno-fluorescence staining of microbial colonization of HHEs cultured for 8 days at the air-liquid interface. *S. aureus* bacteria were exposed to the skin equivalent. Arrows indicate visible bacteria on top of the stratum corneum. **d** Keratinocyte response following bacterial stimulation, qPCR measurement of selected *S. aureus* signature genes.

- In depth analysis of tryptophan (trp) metabolism, where we observed significant upregulation of the kynurenine pathway in the *S. aureus* ‘high’ cohort, and reconstructed trp breakdown in atopic skin inflammation (Fig. 3a), indicating the accumulation of the metabolite 3-hydroxyanthranilic acid (3-HAA), which is considered an inflammatory mediator. Moreover, the depletion of the essential amino acid, trp, may be a mode of host defense during bacterial colonization. Since certain microbes, including staphylococci, are susceptible to the depletion of trp by the host, we investigated whether such mechanisms are at play in AD, during *S. aureus* overcolonization. First we cultured *S. aureus* strains from the skin of an independent group of AD patients, and carried out trp-dependence screening. The results indicate that 66% of isolated *S. aureus* strains were independent of trp (Fig 3c). Second, while we did not have access to bacterial isolates from the MAARS cohort, we examined preliminary whole-genome shotgun metagenomic sequencing (WGS) data, and found that on average, 73% of samples carried members of the trp gene family (Fig. 3d). These results are now included in the manuscript as Fig. S11, and in the text on p.17-18 and discussed on p. 24-25.

Fig. 3. Inference of functional metagenomic features based on 16S rRNA and whole genome sequencing. **a** Regulation of the kynureninase pathway of tryptophan degradation on the mRNA level between HV and the *S. aureus* 'high' cohort. Red arrows: significantly regulated genes, n.s.: not significant. Arrow and font thickness correspond to significance. **b** Contribution of individual OTUs to relevant microbial pathways. The gene content of individual OTUs was inferred using the Greengenes v.13.5 database, and subsequently used to predict enriched microbial pathways in the respective disease groups. Horizontal bars represent the percentage of genes contributed by the most abundant microbes in the dataset. X-axis: sum of relative contributions per sample. Y-axis: contribution across *S. aureus* 'high' and *S. aureus* 'low' cohorts, respectively. **c** Culture-based tryptophan dependence assay of 32 *S. aureus* strains isolated from moderate-to-severe atopic dermatitis patients. Overall, 66% of colonizing *S. aureus* strains were shown to grow independent of tryptophan in Trp-depleted culture medium. **d** Presence of tryptophan biosynthesis-related gene families in WGS sequencing results of *S. aureus* 'high' samples. Y axis: relative abundance of UniRef50-defined *trp* gene families.

4. In order to investigate whether sensitivity to tryptophan depletion is at play in AD during *S. aureus* overcolonization, we sought to identify differences in the metabolic capacity of the microbiome, and generated 16S rRNA-based prediction of the metagenome using the PICRUSt tool (Langille et al., 2013). As a result, we found significantly overrepresented

microbial functions in *S. aureus* ‘high’ compared to ‘low’ groups, such as: ‘bacterial toxins’ (ko02042), ‘phosphotransferase system’ (ko02060), ‘two-component system’ (ko02022), among others (Fig3b, Fig.S11b in the manuscript, Fig.4a). It is notable that due to the great difference in beta diversity, the *S. aureus* ‘high’ cohort represents the bacterial functions of mainly a single species, *S. aureus*, with little contributions from other microbes, while this is contrasted by the functional capacity of a relatively high diversity of microbes in *S. aureus* ‘low’. These results are included in the manuscript as Fig. S11b and Fig.S14a, and in the text on p.17, and discussed on p.24.

- Furthermore, 16S rRNA-based prediction of the metagenome revealed a shift towards glycolysis in the *S. aureus* ‘high’ associated microbiome (Fig. 4a, Fig.S14a in the manuscript). *S. aureus* is known to impose metabolic stress on keratinocytes, resulting in HIF1alpha signaling in the skin, which in turn promotes the generation of inflammatory cytokines, particularly mature IL-1beta (Wickersham et al., 2017). Indeed, in the *S. aureus* ‘high’ samples HIF1a signaling was significantly induced, including HIF1A and HIF1A-dependent genes HK2 and PFKP (Figure 4b, Fig. S14b in the manuscript), and functional analysis predicted IL-1beta as a top upstream regulator, based on the gene expression profiles in the skin (Figure 4d). The results support the notion that metabolic stress, caused by microbial overcolonization as the microorganisms and skin compete for limited oxygen and glucose, drives inflammatory signaling through the induction of HIF1A. These results are included in the manuscript in Fig 14S, and in the text on p. 17 and p. 24-25.

Fig. 4. Prediction of the *S. aureus* metagenome and gene expression in the skin. a Significantly enriched microbial pathways between *S. aureus* ‘high’ and *S. aureus* ‘low’ groups. Gene content of individual OTUs was inferred using the Greengenes v.13.5 database, and subsequently used to predict enriched microbial pathways in the respective disease groups. **b** Expression of HIF1A and HIF1A dependent genes in the skin in the *S. aureus* ‘high’ cohort.

Comment 2: A major point of revision would be to include additional covariate analysis for different skin treatments that the patients had undergone prior to the washout period with Dove, and to describe with more detail the different features of the matched? healthy controls. This is because the authors observed that there were likely different community states/microbiome 'types' for AD because some individuals were characterized by a prevalence of *S. aureus* while others not. Kong et al. reported that intermittent treatment shifts the microbiome towards one with significantly less *S. aureus*, and it is likely that this could be one of the factors that could explain the variation.

ANSWER: The patients were carefully (with strict inclusion and exclusion criteria) selected for this study, omitting those who used systemic antibiotics within 2 weeks, or systemic immunosuppressive therapy, phototherapy, or systemic biologic agents within the previous 12 weeks prior to screening.

We agree this is an important issue to explore, and suggest including it in subsequent studies/analyses of this patient cohort.

Comment 3: Also, why were OTUs clustered at the 99.3% identity level? Which is fairly atypical and naturally resulted in a very large number of OTUs. There should be some justification of this in the methods, and/or also performed at more conventional levels (eg. 97%).

ANSWER: The traditional 97% threshold proposed in 1994 is conservative and could safely be increased, without a significant risk of wrongly differentiated species. In the V4 region, which is one of the two regions (V3-V4) we use in this study, the threshold can be risen much higher, due to stable variable regions as previously described (Meier-Kolthoff JP et al Arch Microbiol. 2013, doi: 10.1007/s00203-013-0888-4).

Importantly, we have now used preliminary data from whole genome sequencing (WGS), to validate 16S based OTU classification and functional predictions based on the 16S marker. The results show close agreement between the two sequencing methods (see Figure 1 above), and are included in the manuscript on p.17 and in Fig. S9D.

Reviewer #2 (Remarks to the Author):

Comment 1: In the paper by Fyhrquist et al., "Microbe-host interplay in atopic dermatitis and psoriasis", the authors assemble a large cohort of cases and controls (over 300 individuals), for which they perform 16S rRNA sequencing and microarray-based analysis of transcriptomes. They find that AD is dominated by *S. aureus* and immune activation, whereas PSO was associated with assemblages of organisms.

I was able to review the technicality of the manuscript as my expertise is in microbial community analysis and gut microbiomes, though I was able to only touch upon the conclusions regarding changes in the immune system pertaining to skin disease and the originality of the work from someone in the microbiome field, but an outsider to these specific areas.

My biggest issue with the manuscript is that it overall lacked details about the study design, sample collection and most importantly, statistical analyses, which made it both hard to review and also gave the impression that many of these choices were made without sufficient consideration. These include the method of DNA extraction (crucial for comparing across studies), the methods for performing skin swabs (Fig. S1 implies some buffer but none was discussed), whether there were any exclusion criteria, how samples were matched, and what Affymetrix plates were used for transcriptomics, to name a few.

ANSWER: This information is available in great detail in the Supplementary material p61-69.

Comment 2: *In terms of statistical analysis, it was unclear why some choices were made—99.3% identity for clustering, how controlling for confounders was done, but also what tests were used (see Figure 1A, a p-value is reported with a vague descriptor of ‘nonparametric score’, but the text only says ‘significantly associated’, yet this repeats in multiple results of the paper, and this is unclear whether this is significantly associated with one of the conditions or if multiple are being compared etc.).*

ANSWER: Concerning 99.3%, please see the answer above.

Confounders were tested and removed as described and illustrated on in the main text on p. 8, in Supplementary material on page 63, last paragraph, in Table S1, and in FigS3C. The analysis of 16S rRNA including the use of statistical tests is described in detail in Supplementary materials, p. 61-64. Differentially abundant OTUs were identified by comparing the abundance distribution of each OTU across the clinical groups (AD, PSO, HV) with the Kruskal-Wallis test (FDR, p-value<0.05). Microbe-disease specific associations were detected testing for the differences in the abundances with the Mann-Whitney U-test (FDR, p-value<0.05).

Comment 3: *Similarly, in the transcriptomics section, the only thing that was mentioned was “Differential gene expression analysis (line 347) rather than the specific method used. There wasn’t mention about false-discovery corrections for some analyses, though mentioned for others (i.e. Fig 4), or whether any normalization was performed on gene abundances. Since the devil is in the details for microbiome analyses, this can potentially be a major issue.*

ANSWER: Affymetrix microarray data were normalized and technical batch effects corrected during preprocessing of the data using state-of-the-art methods, and all critical biological variables were adjusted and tested in the final linear model. Furthermore, Benjamini & Hochberg correction for multiple testing was used throughout the microarray analysis. This information is available in great detail in the Supplementary material p. 65-66.

Comment 4: *My other issue with the manuscript is the novelty of the results. Despite my disclaimer that I don’t work entirely in this field, it seems obvious both from quick lit searches and their own statements that there have been a number of studies that have done community analysis of psoriasis and AD and their analysis doesn’t particularly add much. The S. aureus-association with AD is not a new finding, and this has been reported on many times. There have also been RNA-seq experiments performed on both AD (Suarez-Farinas et al, 2015 and even a metaanalysis of transcriptome analyses (Ding et al., 2015) and PSO (Li et al., 2014). They find many of the same co-occurrences and pathway regulation, even according to their own statements (lines 552, 574). It was hard to decipher whether direct comparison could be made between the two disease cohorts because of the ambiguity of the matching scheme, and also the difference in body sampling sites. I can’t comment whether their results significantly improve or change our understand of PSO or AD.*

ANSWER: We have now added predictions of bacterial pathways, validation of 16S based taxonomy by WGS, functional predictions and mechanistic validations of the observed host-microbe interactions. The analysis brings in several novel aspects of microbe-host-interplay. For more details, please see the answer to reviewer #1 starting on p. 2.

Comment 5: *Other issues:*

- *It was not clear how samples were matched, especially since large differences in gender underlie the Psoriasis and HV cohorts. Similarly, age-related dysbiosis have been reported for skin microbiomes which are thought to predispose individuals to AD. This wasn't tested (or mentioned) within this cohort.*
- *Were there any exclusion criteria? Atopic dermatitis and psoriasis can be treated in many ways, including topical medications (including corticosteroids), oral immunosuppressant therapies, antibiotics, vitamin therapies, and oral steroids as well as immunomodulators. These were never mentioned in the patient recruitment or analysis.*

ANSWER:

Matching of samples: As indicated in Figure S1, the mean ages in the AD, PSO and HV groups are 44.5, 48.8 and 34.9 y, respectively. Women are slightly overrepresented in HV, and men in PSO. To account for these differences, age, gender, anatomical location and clinical center were used as covariates during the extrapolation of differentially expressed genes, and OTUs were corrected for gender, age, anatomical location and clinical center (Table S1). The analysis which combined the transcriptome and the microbiome data, included only individuals which had both types of samples available – a high-quality microbiome sample, and high-quality transcriptome sample.

Exclusion criteria: The exclusion criteria included concomitant autoimmune diseases (e.g. rheumatoid arthritis, diabetes, alopecia areata, etc.) the use of systemic antibiotics within 2 weeks and systemic immunosuppressive therapy or phototherapy or systemic biologic agents within the previous 12 weeks prior to screening. Before skin sampling, the biopsy sites were left untreated for at least 2 weeks and cleansing with only the non-antibacterial Dove soap was allowed and washing was avoided for 24 hours prior to sampling. The patients or healthy volunteers who did not match these clinical exclusion criteria were removed from the study.

This information is available in Supplementary information, p.61.

Minor issues:

Comment 6: - *Hard to follow what is being plotted in many of the figures. I mentioned Fig 1A above, but this is the same for Fig 1B. Is this some aggregate of all of the samples, masking inter-personal variability? This is abnormal to group individual compositions into one composite composition.*

ANSWER: We generated individual compositions of the skin microbiota and included these in the supplementary information (see Fig. 5 below, included in the manuscript as Fig. S2). However, as this study includes 316 samples, we chose to show average microbial compositions of AD, PSO and HV in main Figure 1B in the manuscript.

Comment 7: - *The data display was very confusing—like Figure 3B is very confusing—are these genes up or down in these different conditions and the same direction in these different conditions. This was a confusing figure. Figure 3C/D are also confusing visually. Or what are the yellow edges in Fig 4F?*

ANSWER: The yellow hue inside the dark borders in Fig. 3C and 3D indicate significance, in -log p-values. The yellow edges without a dark border in Fig. 3D indicate log fold change (LogFC). To simplify Fig. 3, we moved panel D to supplementary materials (Figure S7), and included instead dot plots of key genes in HV, AD and PSO.

Fig. 5. Individual compositions of the skin microbiota in AD lesions, HV normal skin and PSO lesions. The most abundant bacterial groups depicted for HV, AD and PSO based on 16S sequences.

Reviewer #3 (Remarks to the Author):

Comment 1: In this manuscript, the authors conducted 16S rRNA sequencing identification of the skin microbiome of patients with atopic dermatitis and psoriasis as well as host transcriptome characterization using microarrays of skin biopsies at the same sites. They showed that *S. aureus* is significantly associated with atopic dermatitis and that multiple species, such as *Corynebacterium* spp and anaerobes are associated with psoriatic skin. They also made classifiers to identify the microbial species that could most differentiate between atopic dermatitis vs normal and psoriasis vs normal skin. They then analyze the host skin transcriptome at these sites.

Many of the findings in this paper confirm previously known studies. The skin microbiome findings for atopic dermatitis reflect what is already known in the literature^{1,2}. Some findings

for the psoriatic skin microbiome are also already known, such as the overrepresentation of *Streptococcus* spp and overabundance of *Corynebacterium* species (reviewed in 3) although this study has a different way of presenting the data (Figures 1-2). The approach of defining microbial classifiers of AD and psoriatic skin is novel, but without any mechanistic data to suggest why particular species in the classifier are disease-defining, the finding has less impact. Additionally, even if species within the classifier are not disease-defining but are in fact secondary effects of a particular type of inflammation, without mechanistic data suggesting why AD vs psoriatic inflammation might bias towards one set of bacteria vs another, it is hard to assess the biological significance of the classifier.

ANSWER: The reviewer raised an important question which unfortunately cannot be addressed here, as it would require a whole new study on its own. However, we believe that our by far largest patient material ever reported in the context of the integration of skin microbiome with the host's cutaneous transcriptome - obtained simultaneously from the same individuals and from the same anatomical locations - provides novel information regarding how microbial communities in the skin are involved in the regulation of skin inflammation in AD and PSO.

Comment 2: The atopic dermatitis transcriptomic data reveal similar pathways as previous skin transcriptome studies^{4,5} and confirms known involved pathways, such as Th1 and Th2 pathways (Figure 3). The psoriasis transcriptomic data also confirm known involved pathways and prior studies (reviewed in 6,7), such as IL-17 signaling and IFN-g signaling (Figure 3). Although Figures 3-4 were clear and beautiful representations of the data, they did not offer new hypothesis-generating insight into disease pathogenesis.

ANSWER: Since the main focus in this paper is on host-microbe-transcriptome interactions, the transcriptomics data *per se* serves mainly as proof of concept.

Comment 3: The main novelty of this study is that skin microbiome and transcriptome data were collected at the same time, to allow for analysis of microbiome-transcriptome correlations. However, in their analysis, the authors did not reveal surprising information. By comparing the transcriptomes in AD samples with high abundance of *S. aureus* to those with low abundance of *S. aureus*, the authors found an enrichment of pathways such as "Keratinization" and enrichment in gene expression of IL18, IL1a, TNF, IFN γ , and other expected pro-inflammatory cytokines (Figure 4C-D).

The analysis of psoriatic microbiomes to transcriptomes revealed that *Corynebacterium* spp may be negatively associated with inflammatory pathways in psoriasis (p.17, line 441-461). However, there are no mechanistic studies to suggest that *Corynebacterium* is protective. Additionally, earlier in the microbiome study, *Corynebacterium* was shown to be overrepresented in psoriatic lesions, which seems contradictory. Perhaps, some *Corynebacterium* species are immunoregulatory or help resistance towards more inflammatory bacteria and are therefore protective, whereas other *Corynebacterium* species are positive associated with psoriasis because they are more pro-inflammatory. This idea would be interesting, but would need additional studies in order to be validated.

There is another finding in the paper that I thought was interesting : abundance of anaerobic bacteria in AD was decreased suggesting that there is increased O₂ tension in AD lesions. (Figure 1D, discussed in line 555-557). An analysis of metabolic pathways upregulated in disease states was a very successful approach to show that utilization of formate and aerobic respiration by *E. coli* was increased in gut inflammation⁽⁸⁾ and then that blocking this particular microbial metabolic pathway using tungsten could ameliorate colitis ⁽⁹⁾. Therefore, if the authors were to use metagenomic sequencing instead of 16s rRNA sequencing to look at the

microbial metabolic pathways that were changed in disease states and correlate this to host transcriptome signatures, this study would be much more informative and more likely to generate hypotheses for further mechanistic studies.

ANSWER: We thank reviewer for this interesting viewpoint. We address now the loss of anaerobes by citing work by Zeeuwen et al (2017) in the manuscript, who introduce a hypothesis regarding the ability of gram-positive anaerobe cocci to induce high levels of antimicrobial peptides in human keratinocytes, thereby strengthening the skin barrier. A complete or partial absence of these organisms, may therefore potentially favor colonization by *S. aureus*. Moreover, we observe metabolic shifts in the AD-associated microbiome (Figures 3-4), generating hypotheses for further mechanistic studies. For further details regarding these, please see above our response to reviewer #1, p. 2.

Comment 4: *The approach and intent of the study was commendable and as a field, we do need to move toward connecting microbiota changes to host epithelial or immunologic changes. However, the data presented here did not reveal many distinctive positive results. If the authors were to re-focus on some of the more interesting findings highlighted above and develop further mechanistic insight instead of presenting a broad overview of findings without specific hypotheses, this would greatly improve the impact of this study.*

ANSWER: We believe that we report here the by far largest patient material ever in the context of integrating the skin microbiome with the host's cutaneous transcriptome in AD and PSO – sampled simultaneously in the same individuals and anatomical locations. Making use of this exceptional resource, we present an overview of our observations, together with novel insights gained from observed correlations between the skin microbiota and host physiology. We have now added validations of 16S based taxonomy by whole genome sequencing (WGS), functional predictions based on the 16S markers, as well as functional insight based on WGS preliminary data. Moreover, we bring in mechanistic validation of the observed host-microbe interactions. Henceforth, this great resource will provide an exceptional asset for continued studies, digging deeper into the interplay between microbial communities in the skin and the regulation of atopic and psoriatic skin inflammation.

References:

1. Byrd, A. L. et al. Staphylococcus aureus and Staphylococcus epidermidis strain diversity underlying pediatric atopic dermatitis. *Sci. Transl. Med.* 9, eaal4651 (2017).
2. Kong, H. H. et al. Temporal shifts in the skin microbiome associated with disease flares and treatment in children with atopic dermatitis. *Genome Res.* 22, 850–859 (2012).
3. Yan, D. et al. The Role of the Skin and Gut Microbiome in Psoriatic Disease. *Curr. Dermatol. Rep.* 6, 94–103 (2017).
4. Ghosh, D. et al. Multiple Transcriptome Data Analysis Reveals Biologically Relevant Atopic Dermatitis Signature Genes and Pathways. *PLOS ONE* 10, e0144316 (2015).
5. Ewald, D. A. et al. Meta-analysis derived atopic dermatitis (MADAD) transcriptome defines a robust AD signature highlighting the involvement of atherosclerosis and lipid metabolism pathways. *BMC Med. Genomics* 8, (2015).
6. Jiang, S., Hincliffe, T. E. & Wu, T. Biomarkers of An Autoimmune Skin Disease—Psoriasis. *Genomics Proteomics Bioinformatics* 13, 224–233 (2015).
7. Lowes, M. A., Suárez-Fariñas, M. & Krueger, J. G. Immunology of Psoriasis. *Annu. Rev. Immunol.* 32, 227–255 (2014).
8. Hughes, E. R. et al. Microbial Respiration and Formate Oxidation as Metabolic Signatures of Inflammation-Associated Dysbiosis. *Cell Host Microbe* 21, 208–219 (2017).
9. Zhu, W. et al. Precision editing of the gut microbiota ameliorates colitis. *Nature* 553, 208–211 (2018).

Reviewers' comments:

Reviewer #1 (Remarks to the Author):

The manuscript has improved, but my impression remains that the value of this manuscript is not from new biological insights, but rather from the size of the dataset investigating simultaneously the microbiome (metagenome) and transcriptome, albeit the microbiome analysis is conducted at the 16S level where metagenomic data would have been desirable. It remains, as I previously noted, a clean, high quality, and well-written study, nonetheless.

One of the potential additional findings of the paper - the link between trp depletion and AD skin response has also been studied (most recently in Dec 2019 the decrease in trp/trp-related metabolites in AD skin was recently published - "A tryptophan metabolite of the skin microbiota attenuates inflammation in atopic dermatitis via the aryl hydrocarbon receptor" that needs to be cited).

The authors did attempt to provide additional data by sequencing 20 of their samples using shotgun metagenomics - how many millions of reads? How many after removal of human DNA? This is important to evaluate the potential depth and whether the trp pathway is actually present or absent. Was trp pathway analysis presence/absence, or can you say something about the relative abundance of the occurrence of the pathway. More generally, the use of PiCrust for strain-specific phenotypes (trp auxotrophy in a portion of strains) is not advised, as in this case, as it can lead to misleading inferences.

I maintain that looking at previous treatment as confounders is important given the high/low *S. aureus* categorization and previous literature on this point. This is a relatively straightforward analysis and there is not really a good reason that it has to be pushed to a second manuscript.

I am not convinced about the rebutting argument to use the 99.4% similarity by looking at the correlation in relative abundance of *S. aureus*. Basically that figure says that the classifications of *S. aureus* are OK (which I agree with), but it remains that the number of OTUs identified is highly inflated. Because the bulk of the analysis doesn't use the data at the OTU level, there is really no value to using this threshold that yields all these OTUs, and I would be sure that many of them are sequencing/technical/artifacts. Phylotyping would be largely fine given the point of the paper. While this is a technical point, it is a trivial one to correct towards a more conservative estimate of microbial diversity.

Reviewer #2 (Remarks to the Author):

I am satisfied with the additional data, information and conclusions in this revised manuscript. The focus on the tryptophan metabolism and the anaerobe discussion improves the unique conclusions of the paper. The inclusion of the metagenomic data is good, although since the PiCRUST stuff is still in there, the authors should show more validation, i.e. check to see if the genes they find in the PiCRUST analysis also correlate in these samples. Even 10 samples of each *S. aureus* high and low cohort, would be nice validation to show. The additional experimental detail is good. I would recommend making sure that the number of samples and the statistical tests used are in the figures. It takes a lot of searching to find this data and makes it hard to interpret at first glance. Another small comment—what immunofluorescence was used in Fig S10C—is the antibody specific to *S. aureus*? (It looks like it could just be DAPI?) I don't think details on the microscopy were included.

Reviewer #3 (Remarks to the Author):

In the revised manuscript, the authors used their transcriptomic data from AD samples with high vs low *S. aureus* abundance to define a host tissue “*S. aureus* signature”. They found the GO term “Tryptophan degradation” to be enriched in the *S. aureus* high group and they also found that *S. aureus* strains in the “high” group seemed to have more tryptophan biosynthesis genes and a majority grew independently of tryptophan.

Here, the authors are trying to draw a more direct connection between their host transcriptome data and their microbe 16s sequencing data, but the logic of this section was confusing to read. Specifically:

- Line 459 – may have a fragment of a sentence or a misplaced comma.
- Line 460-463 – I don't understand how enrichment of microbial pathways like “Phosphotransferase system” etc relate to tryptophan metabolism, which is what this paragraph seems to be mainly about.
- In Figure S11C, Table S4, they found that 66% of isolated *S. aureus* strains grew independently of tryptophan, and I think they are implying that this microbial trait was selected for by decreased host availability of tryptophan due to increased tryptophan metabolism. However, I

don't see in Figures S11C-D a comparison to *S. aureus* "low" AD samples. What if 66% of skin-associated *S. aureus* strains in AD are generally tryptophan-independent, whether or not they exist in a "high *S. aureus*" or "low *S. aureus*" microbial environment?

Additionally, the results regarding upregulated microbial pathways is based on inference from their 16s rRNA sequencing data using PICRUSt. My understanding is that this inference is only as good as the database of sequenced microbial genomes and the ability to match the sequenced strain genomes to the 16s rRNA sequence-generated OTU. Given that different strains of the same skin-colonizing bacterial species can have highly variable behavior in terms of virulence and immune stimulation, that a great number of skin bacterial strains are not sequenced (even if some members of the species have been sequenced), and that 16s rRNA sequencing cannot fully resolve strain-specific differences, how can the authors feel confident in their metagenomic inferences?

I do think, on page 25, the authors present an interesting idea. That is, the host attempt to limit available tryptophan as a defense mechanism against higher *S. aureus* colonization may then cause increased metabolites (such as 3-HAA), which are pro-inflammatory. It is then interesting to wonder if such metabolites like 3-HAA continue to help the host maintain a defensive immune barrier or if they backfire and feed into AD pathology. It is also interesting to wonder if the host defense mechanism of limiting tryptophan is unable to repress *S. aureus* colonization but does in fact repress colonization by benign commensal bacteria, and thus also inadvertently creates more dysbiosis. I wonder if the novelty of these points could be better expressed if the authors condensed lines 658-673 (which seem to mainly restate the results) and highlighted the implications of their data more.

Responses to the reviewers' comments:

Reviewer #1 (Remarks to the Author):

1. *The manuscript has improved, but my impression remains that the value of this manuscript is not from new biological insights, but rather from the size of the dataset investigating simultaneously the microbiome (metagenome) and transcriptome, albeit the microbiome analysis is conducted at the 16S level where metagenomic data would have been desirable. It remains, as I previously noted, a clean, high quality, and well-written study, nonetheless.*

*One of the potential additional findings of the paper - the **link between trp depletion and AD skin response** has also been studied (most recently in Dec 2019 the decrease in trp/trp-related metabolites in AD skin was recently published - "A tryptophan metabolite of the skin microbiota attenuates inflammation in atopic dermatitis via the aryl hydrocarbon receptor" that needs to be cited).*

Was trp pathway analysis presence/absence, or can you say something about the relative abundance of the occurrence of the pathway. More generally, the use of PiCrust for strain-specific phenotypes (trp auxotrophy in a portion of strains) is not advised, as in this case, as it can lead to misleading inferences.

The authors did attempt to provide additional data by sequencing 20 of their samples using shotgun metagenomics - how many millions of reads? How many after removal of human DNA? This is important to evaluate the potential depth and whether the trp pathway is actually present or absent.

Answer: The authors thank the reviewer for referring to the recent study by Yu *et al.* In the revised manuscript, we have now cited the study, please see page 26, lines 698-700. Although in the present work we did not investigate trp metabolites, we maintain that the simultaneous high-throughput characterization of trp metabolism genes in the host transcriptome and microbiome provides novel insights.

Detailed information on sequencing read counts for human and microbial DNA is provided in Table 1 of our point-by-point (please see below). Shotgun metagenomic libraries contained between 2-18.5% microbial DNA while read counts ranged from 0.15-5.5 million and were comparable to those achieved in recent large-scale microbiome studies by Chng *et al* or Tett *et al*^{1,2}. Thus, taking into account the quality of WGS libraries, together with the additional information of independent culture-based Trp-dependence assays (Figure S12b), the authors feel confident in their WGS validation presented in Figure S12c in the manuscript. However, we agree that the role of Trp metabolites in host-microbial interactions in AD is an important issue to further explore in subsequent studies.

Sample ID	Million reads sequenced	Filtered reads	Percentage microbiome
MB1	26.05	725865	2.79
MB2	78.64	2086180	2.65
MB3	54.77	2772356	5.06
MB4	33.03	2373752	7.19
MB5	43.94	1728968	3.93
MB6	29.11	5390992	18.52
MB7	19.94	1354057	6.79
MB8	10.31	208857	2.03
MB9	31.43	2161375	6.88
MB10	18.9	807488	4.27
MB11	52.6	4442216	8.45
MB12	4.51	149775	3.32
MB13	9.38	1053178	11.23
MB14	21.55	823947	3.82
MB15	52.28	1323858	2.53
MB16	30.4	1795493	5.91
MB17	28.23	960793	3.40
MB18	34.17	1408536	4.12
MB19	0.86	136668	15.89
MB20	32.56	2887738	8.87
Average	30.633	1729604.6	6.38

Table 1. Sequence reads before and after removing human DNA.

2. *I maintain that looking at previous treatment as confounders is important given the high/low S. aureus categorization and previous literature on this point. This is a relatively straightforward analysis and there is not really a good reason that it has to be pushed to a second manuscript.*

Answer: We analyzed the potential effect of treatments according to the reviewer suggestions, as shown in Table 2 or our point-by-point reply (please see TABLE 2 at the end of this document). Analysis was carried out in the same manner as confounding factor analysis for age, gender, sampling institution and body site, applying Wilcoxon rank-sum test followed by Benjamini-Hochberg multiple testing correction. We have found no significant associations between treatments and the top 95 most abundant microbiota in atopic dermatitis. In the case of psoriasis, a single *Acinetobacter* OTU showed an association ($p=0.019$) with topical coal tar treatment. Topical coal tar treatment was only used in a small subset within the psoriasis cohort ($n=24$). The specific *Acinetobacter* OTU did not demonstrate any significant associations with the host transcriptome and was not used or referenced in any analyses of the present study. Hence we decided not to include this information into the revised manuscript.

3. *I am not convinced about the rebutting argument to use the 99.4% similarity by looking at the correlation in relative abundance of S. aureus. Basically that figure says that the classifications of S. aureus are OK (which I agree with), but it remains that the number of OTUs identified is highly inflated. Because the bulk of the analysis doesn't use the data at the OTU level, there is really no value to using this threshold that yields all these OTUs, and I would be sure that many of them are sequencing/technical/artifacts. Phylotyping would be largely fine given the point of the paper. While this is a technical point, it is a trivial one to correct towards a more conservative estimate of microbial diversity.*

Answer: We appreciate this concern and we appreciate that the reviewer agrees that the OTU classifications are correct using the higher cutoff. We agree though, that the OTU number we have reported is unnecessarily high as a result. After careful analysis, we have found that the

high number is mainly caused by the generation of a large number of very low-abundance OTUs in our data set, partly caused by the high similarity cutoff. We have therefore solved the problem by filtering out OTUs that have less than 15 reads in the entire dataset. This resulted in a total of **3342 OTUs** instead of the 17k we reported previously. This gives a more accurate representation of the data.

Reviewer #2 (Remarks to the Author):

1. I am satisfied with the additional data, information and conclusions in this revised manuscript. The focus on the tryptophan metabolism and the anaerobe discussion improves the unique conclusions of the paper. The inclusion of the metagenomic data is good, although since the PiCRUST stuff is still in there, the authors should show more validation, i.e. check to see if the genes they find in the PiCRUST analysis also correlate in these samples. Even 10 samples of each *S. aureus* high and low cohort, would be nice validation to show.

Answer: We have analyzed relevant genes as suggested by the Reviewer, validating PiCRUST predictions (Fig. 1, Fig. S8 in the manuscript).

Fig. 1 Prediction and validation of the *S. aureus* metagenome. a Significantly differentially enriched microbial pathways between *S. aureus* 'high' (n=27) and 'low' (n=25) groups. Gene content of individual OTUs was inferred using the Greengenes v.13.5 database, and subsequently used to predict enriched microbial pathways in the respective disease groups. b Abundance of relevant genes by WGS. c Contribution of individual OTUs to relevant microbial pathways. The gene content of individual OTUs was inferred using the Greengenes v.13.5 database, and subsequently used to predict enriched microbial pathways in the respective disease groups. Horizontal bars represent the percentage of genes contributed by the most abundant microbes in the dataset. X-axis: sum of relative contributions per sample. Y-axis: contribution across *S. aureus* 'high' and *S. aureus* 'low' samples, respectively.

2. The additional experimental detail is good. I would recommend making sure that the number of samples and the statistical tests used are in the figures. It takes a lot of searching to find this data and makes it hard to interpret at first glance.

Answer: The details have been added.

3. Another small comment—what immunofluorescence was used in Fig S10C—is the antibody specific to *S. aureus*? (It looks like it could just be DAPI?) I don't think details on the microscopy were included.

Answer: In this experiment, a single *S. aureus* strain (ATCC 29213) was applied onto sterile 3D human epidermal equivalents. We indeed use DAPI (DAPI Fluoromount-G, Southern Biotech) to detect all DNA in the sample, hence the staining of both human DNA (keratinocyte nuclei) and the bacterial DNA on top of the stratum corneum. This staining allows for a visualization of the bacteria to determine the distribution of bacteria topically applied, but does not specifically discriminate bacterial species. Since we in this case applied a single bacterial strain, the DAPI stain herein used is sufficient to discriminate the bacteria in the 3D culture model.

Reviewer #3 (Remarks to the Author):

1. In the revised manuscript, the authors used their transcriptomic data from AD samples with high vs low *S. aureus* abundance to define a host tissue “*S. aureus* signature”. They found the GO term “Tryptophan degradation” to be enriched in the *S. aureus* high group and they also found that *S. aureus* strains in the “high” group seemed to have more tryptophan biosynthesis genes and a majority grew independently of tryptophan.

Here, the authors are trying to draw a more direct connection between their host transcriptome data and their microbe 16s sequencing data, but the logic of this section was confusing to read. Specifically:

- Line 459 – may have a fragment of a sentence or a misplaced comma.

- Line 460-463 – I don't understand how enrichment of microbial pathways like “Phosphotransferase system” etc relate to tryptophan metabolism, which is what this paragraph seems to be mainly about.

Answer: The authors thank the reviewer for this suggestion. The section has been rewritten. Please see page 18, lines 461-470, and page 26-27, lines 696-717. Figures S8 and S12 have been revised accordingly and rearranged for a clearer presentation of the referenced data.

2. - In Figure S11C, Table S4, they found that 66% of isolated *S. aureus* strains grew independently of tryptophan, and I think they are implying that this microbial trait was selected for by decreased host availability of tryptophan due to increased tryptophan metabolism. However, I don't see in Figures S11C-D a comparison to *S. aureus* “low” AD samples. What if 66% of skin-associated *S. aureus* strains in AD are generally tryptophan-independent, whether or not they exist in a “high *S. aureus*” or “low *S. aureus*” microbial environment?

Answer: In *S. aureus* ‘low’ samples, the pathogen is indeed absent, or present at extremely low abundance. Hence, isolation of *S. aureus* strains from these patients is not possible and the coverage of metagenomic shotgun sequencing for the species is also too shallow to identify *S. aureus*-specific Trp gene sequences with the necessary confidence. Trp-dependence assays were performed in *S. aureus* strains isolated from an independent AD collective and findings were confirmed by whole-genome metagenomic sequencing in the MAARS *S. aureus* ‘high’ cohort. On the other hand, assessment of the general independence of *S. aureus* strains from Trp requires analysis and data collection beyond the scope of our present study.

However, the hypothesis that the particular means of host defense by Trp depletion is ineffective against the majority of skin colonizing *S. aureus* strains in the *S. aureus* 'high' group is not affected by these limitations.

3. *Additionally, the results regarding upregulated microbial pathways is based on inference from their 16s rRNA sequencing data using PICRUSt. My understanding is that this inference is only as good as the database of sequenced microbial genomes and the ability to match the sequenced strain genomes to the 16s rRNA sequence-generated OTU. Given that different strains of the same skin-colonizing bacterial species can have highly variable behavior in terms of virulence and immune stimulation, that a great number of skin bacterial strains are not sequenced (even if some members of the species have been sequenced), and that 16s rRNA sequencing cannot fully resolve strain-specific differences, how can the authors feel confident in their metagenomic inferences?*

Answer: Although a significant limitation of prediction methods is, naturally, the missing information on yet unsequenced bacterial strains, the extended ancestral state-reconstruction algorithm developed for PICRUSt has produced up to $r=0.8-0.9$ correlation with WGS datasets³. As our present study focused on a condition dominated by a single, well-studied and well-annotated species, we aimed to gain preliminary insight into the mechanisms of *S. aureus* colonization using the PICRUSt method. Furthermore, we have provided experimental validation of microbial genes predicted to show significantly higher abundance in *S. aureus*-high compared to *S. aureus*-low samples, using preliminary data from 10 whole-genome shotgun sequencing samples of each group. Our validation dataset shows a strikingly high concordance with the findings of 16S rRNA gene-based predictions.

4. *I do think, on page 25, the authors present an interesting idea. That is, the host attempt to limit available tryptophan as a defense mechanism against higher *S. aureus* colonization may then cause increased metabolites (such as 3-HAA), which are pro-inflammatory. It is then interesting to wonder if such metabolites like 3-HAA continue to help the host maintain a defensive immune barrier or if they backfire and feed into AD pathology. It is also interesting to wonder if the host defense mechanism of limiting tryptophan is unable to repress *S. aureus* colonization but does in fact repress colonization by benign commensal bacteria, and thus also inadvertently creates more dysbiosis. I wonder if the novelty of these points could be better expressed if the authors condensed lines 658-673 (which seem to mainly restate the results) and highlighted the implications of their data more.*

Answer: The authors thank the reviewer for this suggestion. The paragraph has been revised accordingly, on page 18, lines 461-470, and page 26-27, lines 696-717. Figures S8 and S12 have been rearranged for a clearer representation of supporting data.

References

- 1 Chng, K. R. *et al.* Whole metagenome profiling reveals skin microbiome-dependent susceptibility to atopic dermatitis flare. *Nat Microbiol* **1**, 16106, doi:10.1038/nmicrobiol.2016.106 (2016).
- 2 Tett, A. *et al.* Unexplored diversity and strain-level structure of the skin microbiome associated with psoriasis. *NPJ Biofilms Microbiomes* **3**, 14, doi:10.1038/s41522-017-0022-5 (2017).
- 3 Langille, M. G. *et al.* Predictive functional profiling of microbial communities using 16S rRNA marker gene sequences. *Nat Biotechnol* **31**, 814-821, doi:10.1038/nbt.2676 (2013).

100	1	Methotrexate	OTU4349859
109	1	Methotrexate	OTU4350124
133	1	Methotrexate	OTU4411187
128	1	Methotrexate	OTU4422405
103	1	Methotrexate	OTU4422718
139	1	Methotrexate	OTU4446521
134.5	1	Methotrexate	OTU4456068
109	1	Methotrexate	OTU4460228
127	1	Methotrexate	OTU4468125
110	1	Methotrexate	OTU4473664
128	1	Methotrexate	OTU4474056
100	1	Methotrexate	OTU4480063
1997,05,01	1	Methotrexate	OTU4482598
124	1	Methotrexate	OTU496787
96	1	Methotrexate	OTU511475
104	1	Methotrexate	OTU610043
110	1	Methotrexate	OTU625320
109	1	Methotrexate	OTU755148
120	1	Methotrexate	OTU761594
103	1	Methotrexate	OTU820692
120	1	Methotrexate	OTU837884
130.5	1	Methotrexate	OTU851668
119	1	Methotrexate	OTU851917
101	1	Methotrexate	OTU851925
106	1	Methotrexate	OTU912906
127	1	Methotrexate	OTU939571
104	1	Methotrexate	OTU940083

580	0.92492995	Fumaricacid	OTU610043
584	0.92492995	Fumaricacid	OTU761594
442	0.92492995	Cyclosporin	OTU114999
876	0.92496879	Retinoids	OTU370134
1942,05,01	0.92496879	Antibiotics	OTU13445
1863	0.92496879	VitaminDanalogs	OTU625320
264	0.92496879	SystemicCorticosteroids	OTU362390
748	0.92496879	Dithranol.minute	OTU1004369
1017	0.92578655	Retinoids	OTU4411187
1191.5	0.92578655	Methotrexate	OTU837884
1182	0.92578655	Methotrexate	OTU851917
1131.5	0.92578655	LocalCorticosteroids	OTU4476950
1276	0.92578655	LocalCorticosteroids	OTU851925
437.5	0.92578655	Topicalcalcineurininhibitors	OTU1131523
345.5	0.92578655	Topicalcalcineurininhibitors	OTU851668
577.5	0.92578655	Fumaricacid	OTU403853
79	0.92578655	Antibiotics	OTU1004369
1864	0.92578655	VitaminDanalogs	OTU1081372
1851	0.92578655	VitaminDanalogs	OTU13445
1026	0.92578655	Photochemotherapy	OTU4349519
1020.5	0.92578655	Photochemotherapy	OTU4480063
1183	0.92578655	Photochemotherapy	OTU610043
1161	0.92578655	Photochemotherapy	OTU883806
266.5	0.92578655	SystemicCorticosteroids	OTU360483
1499.5	0.92578655	Phototherapy	OTU4047452
1701	0.92578655	Phototherapy	OTU4346894
547	0.92578655	Cyclosporin	OTU282360
447.5	0.92578655	Cyclosporin	OTU4350124
756	0.92578655	Dithranol.minute	OTU4317476
754	0.92578655	Dithranol.minute	OTU4474056
446	0.92698449	Cyclosporin	OTU4408996
355	0.9285878	Topicalcalcineurininhibitors	OTU511475
699	0.9285878	Fumaricacid	OTU164003
44	0.9285878	Antibiotics	OTU565753
76	0.9285878	Antibiotics	OTU940083
1070	0.9285878	TAR.topical.	OTU4473201
1178	0.9285878	Photochemotherapy	OTU4460228
194	0.9285878	SystemicCorticosteroids	OTU1081372
192	0.9285878	SystemicCorticosteroids	OTU912997
1511	0.9285878	Phototherapy	OTU4439089
442	0.9285878	Cyclosporin	OTU4440643
444	0.9285878	Cyclosporin	OTU851917
886	0.9285878	Dithranol.minute	OTU279980
762	0.9285878	Dithranol.minute	OTU4476950
1029	0.92881133	Methotrexate	OTU4327300
437	0.92881133	Topicalcalcineurininhibitors	OTU1081372
438	0.92881133	Topicalcalcineurininhibitors	OTU4482598
440	0.92881133	Topicalcalcineurininhibitors	OTU912997
588	0.92881133	Fumaricacid	OTU940083
1944,05,01	0.92881133	Antibiotics	OTU4348347
1944,05,01	0.92881133	Antibiotics	OTU511475
1221	0.92881133	TAR.topical.	OTU912997
1179	0.92881133	Photochemotherapy	OTU4327300
1185.5	0.92881133	Photochemotherapy	OTU837884
442.5	0.92881133	Cyclosporin	OTU4467218
1126.5	0.92938841	LocalCorticosteroids	OTU1096610
347	0.92938841	Topicalcalcineurininhibitors	OTU2110555
696	0.92938841	Fumaricacid	OTU819937
45	0.92938841	Antibiotics	OTU1107940
45	0.92938841	Antibiotics	OTU820692
540	0.92938841	Cyclosporin	OTU4473201
885	0.93132145	Retinoids	OTU4473664
1052	0.93132145	Methotrexate	OTU3841245
1174	0.93132145	Methotrexate	OTU4460228
1034	0.93132145	Methotrexate	OTU610043
1148.5	0.93132145	LocalCorticosteroids	OTU511475
1214	0.93132145	TAR.topical.	OTU4440643
1089	0.93132145	TAR.topical.	OTU883806
1027.5	0.93132145	Photochemotherapy	OTU403853
264	0.93132145	SystemicCorticosteroids	OTU4354809
199	0.93132145	SystemicCorticosteroids	OTU940083
1508	0.93132145	Phototherapy	OTU4449324
1515	0.93132145	Phototherapy	OTU4460228
545	0.93132145	Cyclosporin	OTU4047452
1039	0.93135251	Methotrexate	OTU4021335
1134.5	0.93135251	LocalCorticosteroids	OTU370309
696.5	0.93135251	Fumaricacid	OTU370309
1853.5	0.93135251	VitaminDanalogs	OTU505749
1067	0.93135251	TAR.topical.	OTU2110555
1031	0.93135251	Photochemotherapy	OTU4047452
1035	0.93321	Methotrexate	OTU1036883
1032.5	0.93321	Methotrexate	OTU4369229

1131	0.93321	LocalCorticosteroids	OTU496787
426	0.93321	Topicalcalcineurininhibitors	OTU987144
593	0.93321	Fumaricacid	OTU279980
1858.5	0.93321	VitaminDanalogs	OTU285376
1215	0.93321	TAR.topical.	OTU1004369
1074.5	0.93321	TAR.topical.	OTU247720
771	0.93321	Dithranol.minute	OTU912906
1177.5	0.93459596	Methotrexate	OTU1096610
1136	0.93586675	LocalCorticosteroids	OTU4482598
430	0.93586675	Topicalcalcineurininhibitors	OTU279980
433.5	0.93586675	Topicalcalcineurininhibitors	OTU4301457
1038	0.93586675	Photochemotherapy	OTU4421536
202.5	0.93586675	SystemicCorticosteroids	OTU4468125
1164	0.93621294	Photochemotherapy	OTU4299324
1165	0.93842995	Methotrexate	OTU103810
1844	0.93842995	VitaminDanalogs	OTU114999
1006	0.94041046	Retinoids	OTU4318084
687	0.94041046	Fumaricacid	OTU654307
1204.5	0.94041046	TAR.topical.	OTU819937
891.5	0.94044152	Retinoids	OTU1003210
893.5	0.94044152	Retinoids	OTU164003
890	0.94044152	Retinoids	OTU4422718
892	0.94044152	Retinoids	OTU4439089
894	0.94044152	Retinoids	OTU851917
1170	0.94044152	Methotrexate	OTU370309
1166	0.94044152	Methotrexate	OTU4421536
1272.5	0.94044152	LocalCorticosteroids	OTU4349519
1155	0.94044152	LocalCorticosteroids	OTU4481323
355	0.94044152	Topicalcalcineurininhibitors	OTU20360
430.5	0.94044152	Topicalcalcineurininhibitors	OTU4047452
356	0.94044152	Topicalcalcineurininhibitors	OTU4473664
356	0.94044152	Topicalcalcineurininhibitors	OTU940083
686.5	0.94044152	Fumaricacid	OTU362390
683	0.94044152	Fumaricacid	OTU4299324
687	0.94044152	Fumaricacid	OTU4471315
1948,05,01	0.94044152	Antibiotics	OTU3841245
1841.5	0.94044152	VitaminDanalogs	OTU360483
1828.5	0.94044152	VitaminDanalogs	OTU4348347
1845	0.94044152	VitaminDanalogs	OTU755148
1688.5	0.94044152	VitaminDanalogs	OTU761594
1079	0.94044152	TAR.topical.	OTU4471315
1079.5	0.94044152	TAR.topical.	OTU610043
1085	0.94044152	TAR.topical.	OTU654307
1073.5	0.94044152	TAR.topical.	OTU939571
1164	0.94044152	Photochemotherapy	OTU1131523
1043	0.94044152	Photochemotherapy	OTU939571
205	0.94044152	SystemicCorticosteroids	OTU4294554
205.5	0.94044152	SystemicCorticosteroids	OTU4411187
1525.5	0.94044152	Phototherapy	OTU370309
1535	0.94044152	Phototherapy	OTU378096
1537	0.94044152	Phototherapy	OTU4348347
1678	0.94044152	Phototherapy	OTU4422405
1523.5	0.94044152	Phototherapy	OTU610043
1667	0.94044152	Phototherapy	OTU940083
524.5	0.94044152	Cyclosporin	OTU3841245
771	0.94044152	Dithranol.minute	OTU164003
778	0.94044152	Dithranol.minute	OTU4294554
876	0.94044152	Dithranol.minute	OTU4318084
871	0.94044152	Dithranol.minute	OTU4353642
772	0.94044152	Dithranol.minute	OTU4408996
773	0.94044152	Dithranol.minute	OTU610043
685	0.94088258	Fumaricacid	OTU74351
1675	0.94088258	Phototherapy	OTU995817
1151	0.94113559	Methotrexate	OTU820692
49	0.94113559	Antibiotics	OTU883806
1838	0.94113559	VitaminDanalogs	OTU1003210
1827	0.94113559	VitaminDanalogs	OTU4481323
1080.5	0.94113559	TAR.topical.	OTU505749
1156	0.94113559	Photochemotherapy	OTU4350124
201	0.94113559	SystemicCorticosteroids	OTU285376
259	0.94113559	SystemicCorticosteroids	OTU4422405
453	0.94113559	Cyclosporin	OTU4456068
876	0.94113559	Dithranol.minute	OTU4047452
1666.5	0.9416623	Phototherapy	OTU370134
899	0.94264371	Retinoids	OTU282360
1004	0.94264371	Retinoids	OTU370309
1006	0.94264371	Retinoids	OTU4422405
1162.5	0.94264371	Methotrexate	OTU4047452
1162	0.94264371	Methotrexate	OTU939571
1146	0.94264371	LocalCorticosteroids	OTU912997
1055	0.94264371	Photochemotherapy	OTU4353642
1045	0.94264371	Photochemotherapy	OTU4467218

204.5	0.94264371	SystemicCorticosteroids	OTU164003
204	0.94264371	SystemicCorticosteroids	OTU20360
207	0.94264371	SystemicCorticosteroids	OTU912906
1529	0.94264371	Phototherapy	OTU4482598
455	0.94264371	Cyclosporin	OTU4449324
896.5	0.94388727	Retinoids	OTU4467218
1155	0.94388727	LocalCorticosteroids	OTU4408996
45	0.94388727	Antibiotics	OTU4349859
1826	0.94388727	VitaminDanalogs	OTU912906
1531.5	0.94388727	Phototherapy	OTU4301457
1532.5	0.94388727	Phototherapy	OTU505749
1660.5	0.94390535	Phototherapy	OTU4476950
783	0.94390535	Dithranol.minute	OTU378096
425	0.94438696	Topicalcalcineurininhibitors	OTU4440643
423	0.94438696	Topicalcalcineurininhibitors	OTU625320
205.5	0.94438696	SystemicCorticosteroids	OTU4349519
1264	0.94528897	LocalCorticosteroids	OTU360483
1155	0.94528897	LocalCorticosteroids	OTU819937
1268	0.94574249	LocalCorticosteroids	OTU4449324
254	0.94574249	SystemicCorticosteroids	OTU25259
256	0.94574249	SystemicCorticosteroids	OTU995817
423	0.94596241	Topicalcalcineurininhibitors	OTU4421536
460.5	0.9460888	Cyclosporin	OTU370134
364	0.9465208	Topicalcalcineurininhibitors	OTU14278
422	0.9465208	Topicalcalcineurininhibitors	OTU4439089
530	0.9465208	Cyclosporin	OTU755148
1267	0.94741399	LocalCorticosteroids	OTU403853
1057	0.9479187	Methotrexate	OTU4318084
604	0.9479187	Fumaricacid	OTU851668
1089	0.9479187	TAR.topical.	OTU370134
204.5	0.9479187	SystemicCorticosteroids	OTU837884
1154	0.94834179	Methotrexate	OTU1081372
605	0.94834179	Fumaricacid	OTU4439089
677.5	0.94834179	Fumaricacid	OTU4476950
676	0.94834179	Fumaricacid	OTU912906
1194	0.94834179	TAR.topical.	OTU4422405
1091	0.94834179	TAR.topical.	OTU4473664
1055	0.94834179	Photochemotherapy	OTU755148
1156.5	0.94834179	Photochemotherapy	OTU995817
1651	0.94834179	Phototherapy	OTU13445
1545	0.94834179	Phototherapy	OTU4318084
1660	0.94834179	Phototherapy	OTU4327300
523	0.94834179	Cyclosporin	OTU4348347
458.5	0.94834179	Cyclosporin	OTU496787
522	0.94834179	Cyclosporin	OTU511475
784.5	0.94834179	Dithranol.minute	OTU103810
1700	0.94992251	VitaminDanalogs	OTU4467218
1822	0.94992251	VitaminDanalogs	OTU74351
1153.5	0.94992251	Photochemotherapy	OTU4440643
366.5	0.95089257	Topicalcalcineurininhibitors	OTU4468125
1710	0.95089257	VitaminDanalogs	OTU4299324
1818	0.95089257	VitaminDanalogs	OTU654307
1660	0.95371818	Phototherapy	OTU1004369
70	0.95691009	Antibiotics	OTU610043
1711	0.95691009	VitaminDanalogs	OTU362390
915.5	0.95725431	Retinoids	OTU4348347
909	0.95725431	Retinoids	OTU625320
607.5	0.95725431	Fumaricacid	OTU360483
1708.5	0.95725431	VitaminDanalogs	OTU819937
211	0.95725431	SystemicCorticosteroids	OTU14278
251	0.95725431	SystemicCorticosteroids	OTU4471315
792	0.95725431	Dithranol.minute	OTU1107940
1103	0.95854109	TAR.topical.	OTU1107940
991	0.95974721	Retinoids	OTU20360
907.5	0.95974721	Retinoids	OTU403853
1146	0.95974721	Methotrexate	OTU4303697
1249	0.95974721	LocalCorticosteroids	OTU362390
368	0.95974721	Topicalcalcineurininhibitors	OTU654307
614	0.95974721	Fumaricacid	OTU4481323
70	0.95974721	Antibiotics	OTU4449324
1097	0.95974721	TAR.topical.	OTU114999
1094.5	0.95974721	TAR.topical.	OTU4449324
1063.5	0.95974721	Photochemotherapy	OTU103810
1066	0.95974721	Photochemotherapy	OTU14278
1061	0.95974721	Photochemotherapy	OTU164003
1143	0.95974721	Photochemotherapy	OTU205025
1148	0.95974721	Photochemotherapy	OTU2110555
1147	0.95974721	Photochemotherapy	OTU25259
249	0.95974721	SystemicCorticosteroids	OTU4303697
1549	0.95974721	Phototherapy	OTU4303697
1650	0.95974721	Phototherapy	OTU4306540
466.5	0.95974721	Cyclosporin	OTU4349519

784.5	0.95974721	Dithranol.minute	OTU4440643
367	0.9601394	Topicalcalcineurininhibitors	OTU4327286
1185.5	0.9601394	TAR.topical.	OTU403853
1067	0.9609343	Methotrexate	OTU14278
1146	0.9609343	Methotrexate	OTU4474056
1252	0.9609343	LocalCorticosteroids	OTU1036883
1651.5	0.9609343	Phototherapy	OTU4456068
523	0.9609343	Cyclosporin	OTU2110555
524.5	0.9609343	Cyclosporin	OTU403853
521.5	0.9609343	Cyclosporin	OTU851668
787	0.9609343	Dithranol.minute	OTU4354809
1147.5	0.96113607	Methotrexate	OTU4349859
1800	0.96113607	VitaminDanalogs	OTU883806
1141	0.9616637	Methotrexate	OTU114999
1065.5	0.9616637	Methotrexate	OTU4349519
1066	0.9616637	Methotrexate	OTU74351
1175	0.9616637	LocalCorticosteroids	OTU13445
1169	0.9616637	LocalCorticosteroids	OTU755148
1252.5	0.9616637	LocalCorticosteroids	OTU995817
666	0.9616637	Fumaricacid	OTU565753
675	0.9616637	Fumaricacid	OTU837884
1807.5	0.9616637	VitaminDanalogs	OTU4350124
1177.5	0.9616637	TAR.topical.	OTU25478
1102	0.9616637	TAR.topical.	OTU4303697
1648	0.9616637	Phototherapy	OTU4354809
1644	0.9616637	Phototherapy	OTU4422718
1640	0.9616637	Phototherapy	OTU4481323
520	0.9616637	Cyclosporin	OTU4303697
471	0.9616637	Cyclosporin	OTU761594
792	0.9616637	Dithranol.minute	OTU4303697
918.5	0.96259709	Retinoids	OTU761594
416	0.96259709	Topicalcalcineurininhibitors	OTU285376
1066.5	0.96259709	Photochemotherapy	OTU20360
244	0.96259709	SystemicCorticosteroids	OTU820692
1643	0.96259709	Phototherapy	OTU282360
519.5	0.96259709	Cyclosporin	OTU164003
668.5	0.96330929	Fumaricacid	OTU114999
1069.5	0.96330929	Photochemotherapy	OTU4476950
854.5	0.96330929	Dithranol.minute	OTU1131523
856.5	0.96330929	Dithranol.minute	OTU4369229
68	0.96377545	Antibiotics	OTU4369229
1642	0.96377545	Phototherapy	OTU851917
1641	0.96422706	Phototherapy	OTU25259
1070	0.9664714	Photochemotherapy	OTU819937
413	0.96714331	Topicalcalcineurininhibitors	OTU755148
979	0.96726303	Retinoids	OTU4309323
916	0.96726303	Retinoids	OTU912997
618	0.96726303	Fumaricacid	OTU103810
68	0.96726303	Antibiotics	OTU285376
243	0.96726303	SystemicCorticosteroids	OTU1107940
1557	0.96726303	Phototherapy	OTU939571
980	0.96861598	Retinoids	OTU360483
928	0.96861598	Retinoids	OTU3841245
981	0.96861598	Retinoids	OTU4354809
1072	0.96861598	Methotrexate	OTU4446521
1132	0.96861598	Methotrexate	OTU4481323
410	0.96861598	Topicalcalcineurininhibitors	OTU362390
618	0.96861598	Fumaricacid	OTU4473664
662	0.96861598	Fumaricacid	OTU820692
1804	0.96861598	VitaminDanalogs	OTU4471315
1107	0.96861598	TAR.topical.	OTU755148
241	0.96861598	SystemicCorticosteroids	OTU883806
516	0.96861598	Cyclosporin	OTU4421536
516	0.96861598	Cyclosporin	OTU625320
516	0.96861598	Cyclosporin	OTU819937
796	0.96861598	Dithranol.minute	OTU2110555
797	0.96861598	Dithranol.minute	OTU25259
1171	0.97052901	TAR.topical.	OTU282360
979	0.97056003	Retinoids	OTU1004369
923	0.97056003	Retinoids	OTU2110555
925	0.97056003	Retinoids	OTU247720
976	0.97056003	Retinoids	OTU4021335
977	0.97056003	Retinoids	OTU4471315
930	0.97056003	Retinoids	OTU883806
1074	0.97056003	Methotrexate	OTU370134
1183	0.97056003	LocalCorticosteroids	OTU565753
664	0.97056003	Fumaricacid	OTU4303697
1802	0.97056003	VitaminDanalogs	OTU2110555
1133.5	0.97056003	Photochemotherapy	OTU370309
1074	0.97056003	Photochemotherapy	OTU4482598
243	0.97056003	SystemicCorticosteroids	OTU1131523
243	0.97056003	SystemicCorticosteroids	OTU282360

218	0.97056003	SystemicCorticosteroids	OTU3208510
1632	0.97056003	Phototherapy	OTU114999
1634	0.97056003	Phototherapy	OTU279980
1563	0.97056003	Phototherapy	OTU755148
515	0.97056003	Cyclosporin	OTU4439089
845	0.97056003	Dithranol.minute	OTU851925
927	0.97123767	Retinoids	OTU3208510
924.5	0.97123767	Retinoids	OTU4306540
1180	0.97123767	LocalCorticosteroids	OTU2901965
1184	0.97123767	LocalCorticosteroids	OTU378096
623.5	0.97123767	Fumaricacid	OTU4350124
619	0.97123767	Fumaricacid	OTU912997
1736	0.97123767	VitaminDanalogs	OTU565753
1623	0.97123767	Phototherapy	OTU3841245
1635	0.97123767	Phototherapy	OTU496787
846.5	0.97123767	Dithranol.minute	OTU819937
798.5	0.97123767	Dithranol.minute	OTU837884
1130	0.97270009	Photochemotherapy	OTU4446521
1129.5	0.97339847	Photochemotherapy	OTU282360
242	0.97339847	SystemicCorticosteroids	OTU4473664
217.5	0.97405353	SystemicCorticosteroids	OTU4369229
1233	0.97418992	LocalCorticosteroids	OTU14278
622	0.97418992	Fumaricacid	OTU4327300
1081	0.97433502	Photochemotherapy	OTU362390
1127	0.97825737	Photochemotherapy	OTU247720
1082	0.97844062	Photochemotherapy	OTU3208510
378.5	0.97854183	Topicalcalcineurininhibitors	OTU103810
1166.5	0.97854183	TAR.topical.	OTU496787
1235	0.97964384	LocalCorticosteroids	OTU4354809
626.5	0.97964384	Fumaricacid	OTU13445
1118	0.97964384	TAR.topical.	OTU761594
479	0.97964384	Cyclosporin	OTU4317476
929	0.97977109	Retinoids	OTU4303697
1738.5	0.97977109	VitaminDanalogs	OTU378096
1788	0.97999566	VitaminDanalogs	OTU987144
927.5	0.98135141	Retinoids	OTU4346894
403	0.98135141	Topicalcalcineurininhibitors	OTU851925
479.5	0.98135141	Cyclosporin	OTU2901965
221	0.98248499	SystemicCorticosteroids	OTU4481323
1083	0.98365272	Methotrexate	OTU25259
1119	0.98365272	Methotrexate	OTU883806
1083.5	0.98365272	Photochemotherapy	OTU4349522
1126	0.98365272	Photochemotherapy	OTU4474056
1621.5	0.98365272	Phototherapy	OTU4468125
509	0.98365272	Cyclosporin	OTU4460228
1124	0.98378876	Methotrexate	OTU4440643
1122	0.98378876	Methotrexate	OTU4473201
1120.5	0.98378876	Methotrexate	OTU511475
657	0.98378876	Fumaricacid	OTU4317476
1741	0.98378876	VitaminDanalogs	OTU4021335
1119	0.98378876	TAR.topical.	OTU4467218
1084	0.98378876	Photochemotherapy	OTU625320
1623.5	0.98378876	Phototherapy	OTU4467218
1619	0.98378876	Phototherapy	OTU565753
1576	0.98378876	Phototherapy	OTU851668
508	0.98378876	Cyclosporin	OTU4471315
841	0.98378876	Dithranol.minute	OTU1036883
509	0.98448982	Cyclosporin	OTU837884
934.5	0.98530666	Retinoids	OTU103810
966	0.98530666	Retinoids	OTU4446521
932.5	0.98530666	Retinoids	OTU496787
656	0.98530666	Fumaricacid	OTU2110555
653	0.98530666	Fumaricacid	OTU851925
1785.5	0.98530666	VitaminDanalogs	OTU4476950
1158	0.98530666	TAR.topical.	OTU360483
1579	0.98530666	Phototherapy	OTU4473201
1618	0.98530666	Phototherapy	OTU912906
508	0.98530666	Cyclosporin	OTU4354809
485	0.98530666	Cyclosporin	OTU820692
965	0.987455	Retinoids	OTU114999
1121	0.987455	Photochemotherapy	OTU1036883
1619.5	0.9874968	Phototherapy	OTU4440643
937.5	0.98835338	Retinoids	OTU511475
1125.5	0.98835338	TAR.topical.	OTU378096
631	0.98847033	Fumaricacid	OTU3208510
1117	0.98847033	Photochemotherapy	OTU820692
810	0.98847033	Dithranol.minute	OTU2901965
937	0.98891134	Retinoids	OTU74351
1194	0.98891134	LocalCorticosteroids	OTU4318084
630	0.98891134	Fumaricacid	OTU4474056
63	0.98891134	Antibiotics	OTU20360
63	0.98891134	Antibiotics	OTU4301457

1153	0.98891134	TAR.topical.	OTU4294554
1118	0.98891134	Photochemotherapy	OTU4021335
1088	0.98891134	Photochemotherapy	OTU4422405
653	0.98987863	Fumaricacid	OTU2901965
652	0.98987863	Fumaricacid	OTU4309323
1126	0.98987863	TAR.topical.	OTU851668
812	0.99051617	Dithranol.minute	OTU1003210
1116	0.99067792	Methotrexate	OTU4299324
1195	0.99067792	LocalCorticosteroids	OTU4471315
385	0.99067792	Topicalcalcineurininhibitors	OTU4481323
1613	0.99067792	Phototherapy	OTU14278
1583	0.99067792	Phototherapy	OTU4474056
811.5	0.99067792	Dithranol.minute	OTU4449324
651	0.99146669	Fumaricacid	OTU4473201
503.5	0.99146669	Cyclosporin	OTU610043
1152	0.99172584	TAR.topical.	OTU940083
812.5	0.99172584	Dithranol.minute	OTU496787
939	0.99373266	Retinoids	OTU4317476
1779	0.99373266	VitaminDanalogs	OTU939571
1116	0.99465828	Methotrexate	OTU4422405
1115	0.99465828	Methotrexate	OTU505749
502	0.99465828	Cyclosporin	OTU3208510
815.5	0.99465828	Dithranol.minute	OTU4411187
815	0.99465828	Dithranol.minute	OTU761594
1752	0.9949947	VitaminDanalogs	OTU3208510
1114	0.99621042	Methotrexate	OTU4467218
1201	0.99621042	LocalCorticosteroids	OTU1107940
1149	0.99621042	TAR.topical.	OTU4439089
1112.5	0.99621042	Photochemotherapy	OTU4318084
234	0.99621042	SystemicCorticosteroids	OTU851925
1610	0.99621042	Phototherapy	OTU164003
1611	0.99621042	Phototherapy	OTU285376
1609	0.99621042	Phototherapy	OTU3208510
501.5	0.99621042	Cyclosporin	OTU20360
1610	0.99797008	Phototherapy	OTU403853
1590.5	0.99797008	Phototherapy	OTU511475
1095	0.99800556	Methotrexate	OTU4449324
949.5	1	Retinoids	OTU285376
1110	1	Methotrexate	OTU4349522
1111	1	Methotrexate	OTU4482598
1110	1	Methotrexate	OTU940083
1211.5	1	LocalCorticosteroids	OTU370134
1208	1	LocalCorticosteroids	OTU4446521
1209	1	LocalCorticosteroids	OTU912906
389.5	1	Topicalcalcineurininhibitors	OTU13445
390	1	Topicalcalcineurininhibitors	OTU4353642
393	1	Topicalcalcineurininhibitors	OTU4446521
391	1	Topicalcalcineurininhibitors	OTU4460228
393	1	Topicalcalcineurininhibitors	OTU505749
388	1	Topicalcalcineurininhibitors	OTU837884
395.5	1	Topicalcalcineurininhibitors	OTU939571
638	1	Fumaricacid	OTU1004369
639	1	Fumaricacid	OTU4408996
638	1	Fumaricacid	OTU4482598
642	1	Fumaricacid	OTU939571
1760.5	1	VitaminDanalogs	OTU20360
1766	1	VitaminDanalogs	OTU247720
1762.5	1	VitaminDanalogs	OTU2901965
1761	1	VitaminDanalogs	OTU370309
1770	1	VitaminDanalogs	OTU4327300
1769	1	VitaminDanalogs	OTU4421536
1755	1	VitaminDanalogs	OTU4449324
1146	1	TAR.topical.	OTU14278
1145	1	TAR.topical.	OTU4317476
1135	1	TAR.topical.	OTU4460228
1138	1	TAR.topical.	OTU4474056
1139	1	TAR.topical.	OTU511475
1098	1	Photochemotherapy	OTU1107940
1103.5	1	Photochemotherapy	OTU13445
1105	1	Photochemotherapy	OTU2901965
1107	1	Photochemotherapy	OTU4294554
1108	1	Photochemotherapy	OTU4473201
1100	1	Photochemotherapy	OTU505749
1100	1	Photochemotherapy	OTU511475
1099	1	Photochemotherapy	OTU851925
230	1	SystemicCorticosteroids	OTU4327300
233.5	1	SystemicCorticosteroids	OTU4350124
229	1	SystemicCorticosteroids	OTU4353642
229.5	1	SystemicCorticosteroids	OTU511475
227	1	SystemicCorticosteroids	OTU565753
230	1	SystemicCorticosteroids	OTU819937
1596	1	Phototherapy	OTU360483

1592	1	Phototherapy	OTU4021335
1602	1	Phototherapy	OTU4350124
494.5	1	Cyclosporin	OTU285376
499	1	Cyclosporin	OTU360483
498	1	Cyclosporin	OTU4021335
498	1	Cyclosporin	OTU4369229
494	1	Cyclosporin	OTU4411187
495	1	Cyclosporin	OTU4468125
818.5	1	Dithranol.minute	OTU114999
823	1	Dithranol.minute	OTU4348347

REVIEWERS' COMMENTS:

Reviewer #1 (Remarks to the Author):

We feel the authors have sufficiently addressed my concerns for the manuscript. Congratulations on a great paper.

Reviewer #2 (Remarks to the Author):

I am satisfied with what the authors have done to address the comments.

Reviewer #3 (Remarks to the Author):

The manuscript has improved; the sections in the results and discussion regarding interpretation of the *S. aureus* and host transcriptional signatures is now much better explained.

For Figure S8b, I could not find the methods about how you picked the genes shown in Figure S8b. For example, under "Two_Component_System", only 15 genes have abundance data shown. Were these ones picked because they showed the best differences across the SA high vs low categories? Or were these the ones that had the best WGS coverage? I'm sorry if the methods are clearly stated somewhere, but I couldn't find it easily.

Responses to the reviewers' comments:

Reviewer #3 (remarks to the author):

For Figure S8b, I could not find the methods about how you picked the genes shown in Figure S8b. For example, under "Two_Component_System", only 15 genes have abundance data shown. Were these ones picked because they showed the best differences across the SA high vs low categories? Or were these the ones that had the best WGS coverage? I'm sorry if the methods are clearly stated somewhere, but I couldn't find it easily.

Answer:

Indeed, the heatmap presented in Figure S8b shows the top most significantly differentially abundant set of microbial genes between the *S. aureus*-high and *S. aureus*-low, with a cutoff of log₂ fold-change > 1 and adjusted p-value < 0.05, annotated with the relevant KEGG pathway categories. The statistical test used was the zero-inflated Gaussian model implemented in the metagenomeSeq R package with adjustment for sampling site, body site, age, sex and library preparation. The figure legend has been updated accordingly (Supplementary Fig. 8b).